# WHAT MAKES A MAZE LOOK LIKE A MAZE?

**Joy Hsu**
Stanford University
joycj@stanford.edu

**Jiayuan Mao**
MIT
jiayuanm@mit.edu

**Joshua B. Tenenbaum**
MIT
jbt@mit.edu

**Noah D. Goodman**
Stanford University
ngoodman@stanford.edu

**Jiajun Wu**
Stanford University
jiajunwu@cs.stanford.edu

## ABSTRACT

A unique aspect of human visual understanding is the ability to flexibly interpret abstract concepts: acquiring lifted rules explaining what they symbolize, grounding them across familiar and unfamiliar contexts, and making predictions or reasoning about them. While off-the-shelf vision-language models excel at making literal interpretations of images (e.g., recognizing object categories such as tree branches), they still struggle to make sense of such visual abstractions (e.g., how an arrangement of tree branches may form the walls of a maze). To address this challenge, we introduce Deep Schema Grounding (DSG), a framework that leverages explicit structured representations of visual abstractions for grounding and reasoning. At the core of DSG are *schemas*—dependency graph descriptions of abstract concepts that decompose them into more primitive-level symbols. DSG uses large language models to extract schemas, then hierarchically grounds concrete to abstract components of the schema onto images with vision-language models. The grounded schema is used to augment visual abstraction understanding. We systematically evaluate DSG and different methods in reasoning on our new Visual Abstractions Benchmark, which consists of diverse, real-world images of abstract concepts and corresponding question-answer pairs labeled by humans. We show that DSG significantly improves the abstract visual reasoning performance of vision-language models, and is a step toward human-aligned understanding of visual abstractions.

## 1 INTRODUCTION

Humans possess the remarkable ability to flexibly acquire and apply abstract concepts when interpreting the concrete world around us. Consider the concept "maze": our mental model can interpret mazes constructed with conventional materials (e.g., drawn lines) or unconventional ones (e.g., icing), and reason about mazes across a wide range of configurations and environments (e.g., in a cardboard box or on a knitted square). Our goal is to build systems that can make such flexible and broad generalizations as humans do. This necessitates a reconsideration of a fundamental question: *what makes a maze look like a maze?* A maze is not defined by concrete visual features such as the specific material of walls or particular perpendicular intersections, but by lifted rules over symbols—a plausible model for a maze includes its layout, the walls, and the designated entry and exit. Crucially, lifted model components can be realized by infinitely many real-world variations, as opposed to grounded concepts which are tied to specific visual inputs. For instance, walls constructed from candy canes, hay, or popsicle sticks, despite their diverse materials, are all identifiable as parts of a maze.

However, current vision-language models (VLMs) often struggle to reason about visual abstractions at a human level, frequently defaulting to literal interpretations of images, such as a collection of object categories. These interpretations may be technically correct descriptions of the scene, but may not align with the abstract concept underlying the image. For example, when given the intended concept of "maze" and an image that realizes the concept with unconventional objects, VLMs such as GPT-4o (OpenAI, 2024) and LLaVA (Liu et al., 2024) fail to correctly recognize objects in the scene as components of the visual abstraction. It is unclear whether these off-the-shelf models leverage knowledge of abstract concepts to make sense of images as humans do.

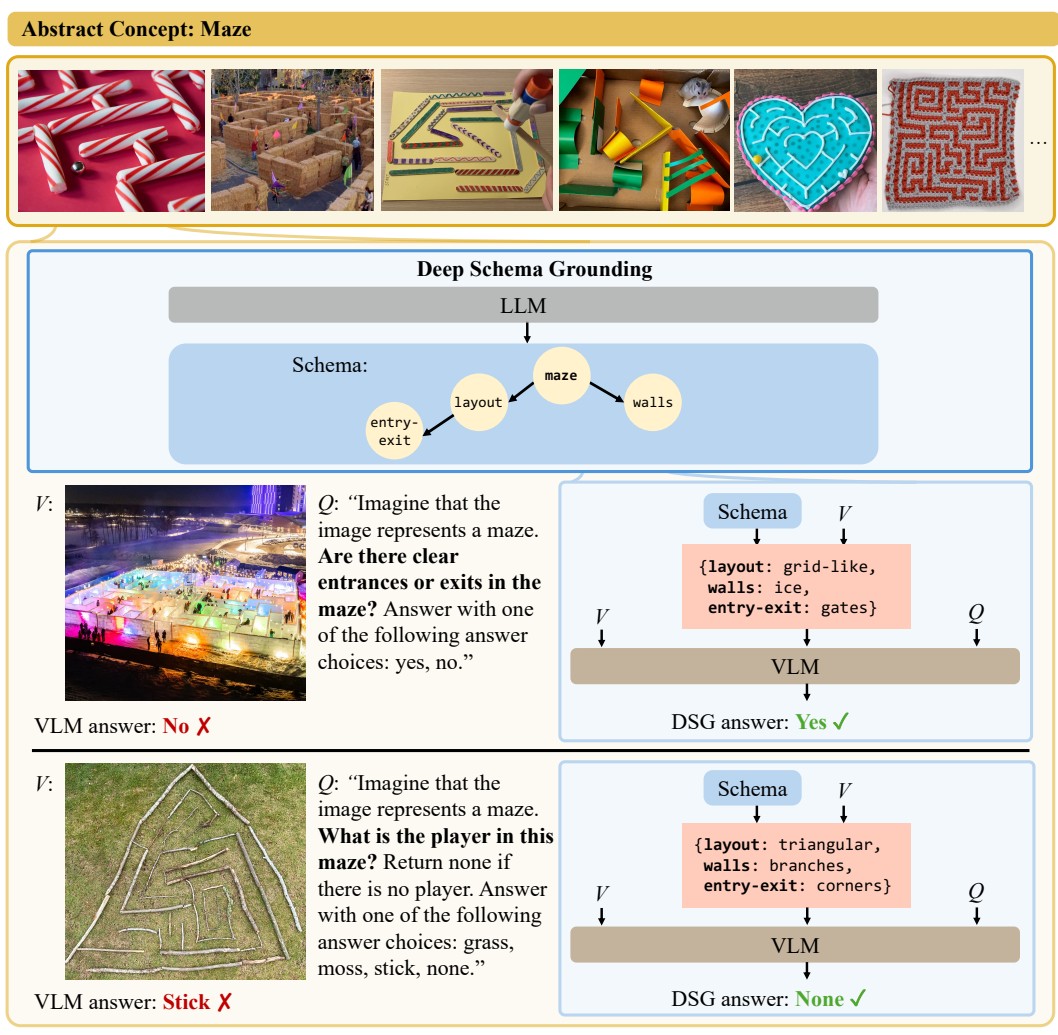

Figure 1: There exist abstract concepts, such as "maze", which are defined by lifted symbols and patterns, instead of concrete visual features. We propose Deep Schema Grounding (DSG), a framework for visual reasoning over such abstract concepts, which uses *schemas* to structure models' interpretation of images. DSG hierarchically grounds conceptual schemas on images and uses them to provide holistic context to VLMs, improving performance across diverse downstream queries.

To address the challenge of understanding abstract visual concepts, we propose Deep Schema Grounding (DSG), a framework for models to interpret visual abstractions. At the core of DSG are *schemas*—a dependency graph description of abstract concepts (see Figure 1). Schemas characterize common patterns that humans use to interpret the visual world, generalize efficiently from limited data, and reason across multiple levels of abstraction for flexible adaptation (Schank & Abelson, 1975). A schema for "helping" allows us to understand relations between characters in a finger puppet scene, while a schema for "tic-tac-toe" allows us to play the game even when the grid is composed of hula hoops instead of drawn lines. A schema for "maze" makes a maze look like a maze.

DSG explicitly uses schemas generated by and grounded by large pretrained models to reason about visual abstractions. Concretely, we model schemas as programs encoding directed acyclic graphs (DAGs), which decompose an abstract concept into a set of more concrete visual concepts as subcomponents, as illustrated in Figure 1. The full framework is composed of three steps. First, we extract schema definitions of abstract concepts from a large language model (LLM). Next, DSG hierarchically queries a vision-language model (VLM), first grounding concrete symbols in the DAG (i.e. symbols that do not depend on the interpretation of other symbols), then using those symbols as conditions to ground more abstract symbols. Finally, we use the resolved schema, including the grounding of all its components, as an additional input into a vision-language model to provide holistic context about the image, such that we can improve visual reasoning across diverse downstream

queries about the abstract concept. Our method is a general framework for abstract concepts without dependency on specific models; the LLMs and VLMs used are interchangeable.

To investigate the capabilities of models in understanding visual abstractions, we introduce the Visual Abstractions Benchmark (VAB). VAB is a visual question-answering benchmark that consists of diverse, real-world images representing abstract concepts. The abstract concepts span 4 different categories: strategic concepts that are characterized by rules and patterns (e.g., "tic-tac-toe"), scientific concepts of phenomena that cannot be visualized in their canonical forms (e.g., "cell"), social concepts that are defined by theory-of-mind relations (e.g., "deceiving"), and domestic concepts of household objectives that cannot be directly defined by specific arrangements of objects (e.g, "table setting for two"). Each image is an instantiation of an abstract concept, and is paired with questions that probe understanding of the visual abstraction; for example, "Imagine that the image represents a maze. Are there clear entrances or exits in the maze?" The Visual Abstractions Benchmark comprises 540 of such examples, with answers labeled by 5 human annotators from Prolific. We found that off-the-shelf VLMs and integrated LLMs with APIs have much room for improvement on this benchmark.

We evaluate Deep Schema Grounding on the Visual Abstractions Benchmark, and show that DSG consistently improves performance of vision-language models across question types, abstract concept categories, and base models. Notably, DSG improves GPT-4o by 6.6 percent points overall ($\uparrow$ 9.9% relative improvement), and, in particular, demonstrates a 10 percent point improvement ($\uparrow$ 16.6% relative improvement) in questions that involve counting. While the challenge of visual abstraction understanding is still far from being fully solved, our results show that DSG is a promising solution that effectively uses schemas to structure the thinking process of visual reasoning systems.

In summary, our contributions are the following:

- We propose Deep Schema Grounding (DSG), a hierarchical, decomposition-based framework that explicitly extracts and grounds schemas of concepts with large pretrained models.
- We introduce the Visual Abstractions Benchmark (VAB), a diverse, real-world visual question-answering benchmark that evaluates VLMs' understanding of visual abstractions.
- We demonstrate DSG's improvement over prior state-of-the-art works across a variety of abstract concept categories, question types, response types, and metrics.

## 2 RELATED WORKS

**Vision-language grounding and reasoning benchmarks.** Visual-language benchmarks have traditionally tackled reasoning about content in images, such as objects, relations, and actions (Chen et al., 2015; Yu et al., 2015; Antol et al., 2015; Wu et al., 2016; Zhu et al., 2016; Krishna et al., 2017; Wang et al., 2017). Recently, an increasing number of works have focused on evaluating visual-language tasks that require commonsense reasoning, usually by asking "what if" or "why" questions (Pirsiavash et al., 2014; Wagner et al., 2018; Zellers et al., 2019; Lu et al., 2022), or by generating holistic scene descriptions (Song et al., 2024). In contrast to these related works, our paper emphasizes the visual grounding and reasoning problem of *abstract* concepts, such as uncommon visualizations of scientific terms or unconventional constructions of strategic games. Such abstract concepts should be grounded on the relationship between lower-level concepts such as entities and patterns, therefore positing novel challenges to vision-language models.

**Concept representations in minds and machines.** Our schema-based concept representations are inspired by the theory-theory of concepts in cognitive science and artificial intelligence (Schank & Abelson, 1975; Morton, 1980; Carey, 1985; Gopnik, 1988; Gopnik & Meltzoff, 1997; Carey, 2000). In short, these works treat concepts as organized within and around theories. Therefore, acquiring the concept involves learning its theory, and reasoning with a concept involves causal-explanatory reasoning of the theories. Such views have been largely leveraged in generating symbolic representations for concepts such as word meanings (Speer et al., 2017) and object shapes (Biederman, 1987). However, most works have focused on using only symbolic features to describe scenes and have been primarily applied to simple object-level concrete concepts. In contrast, we propose using our schema representation as a guide for grounding scene-level, abstract concepts in natural images.

**Hierarchical grounding of visual concepts.** Our approach is related to recent works on chain-of-thought reasoning in vision-language models (Lu et al., 2022; Chen et al., 2024), hierarchical image classification (Koh et al., 2020; Menon & Vondrick, 2023), hierarchical relation detection (Li et al., 2024), and visual reasoning via programs (Hu et al., 2017; Mao et al., 2019; Hsu et al., 2024). In contrast to these algorithms, which are primarily designed for object and relational reasoning,

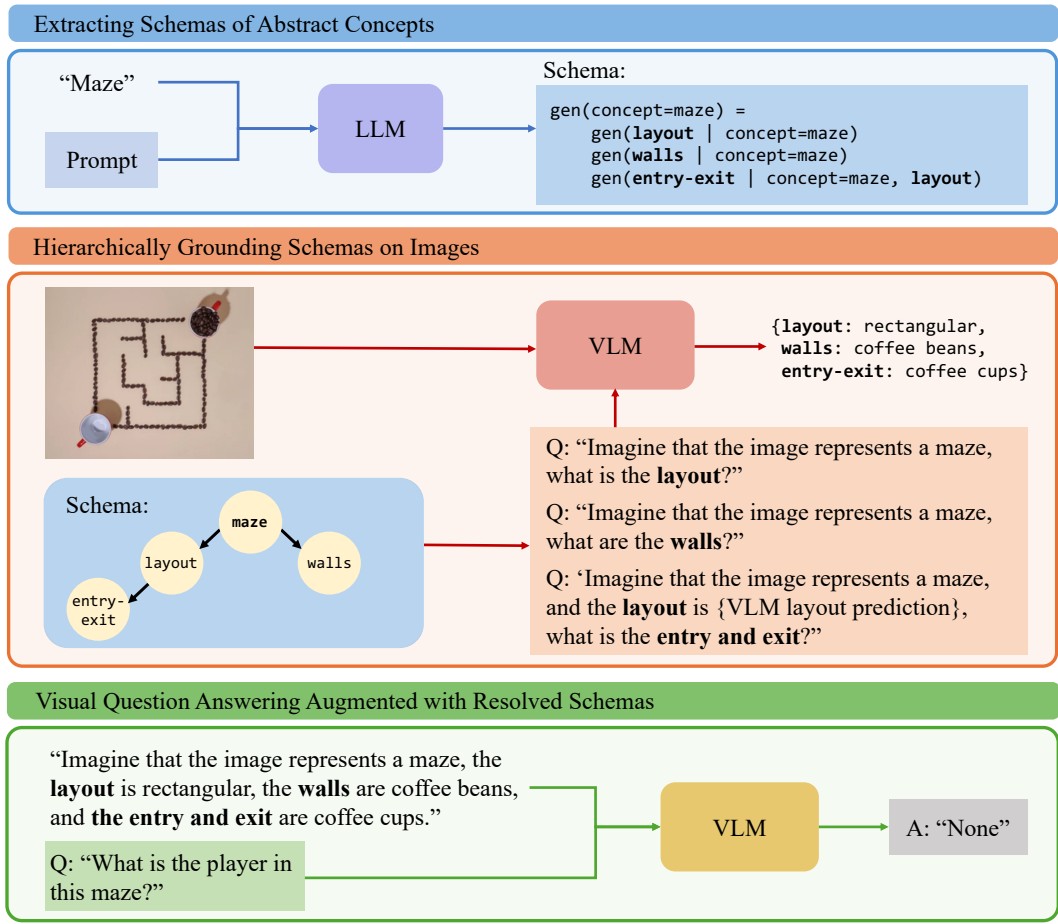

Figure 2: Deep Schema Grounding consists of three main stages: (1) extracting schemas of abstract concepts with large language models, (2) hierarchically grounding schemas on images with vision-language models, and (3) conducting visual question-answering augmented with resolved schemas.

we focus on the grounding of scene-level concepts. Our framework also shares a similar spirit to prior approaches (Menon & Vondrick, 2023; Li et al., 2024) that leverage vision-language models to generate text descriptions of class features for classification tasks. However, our paper proposes the generation of hierarchical schemas with LLMs to ground visual abstractions for reasoning. Additionally, our hierarchical schema representation is related to hierarchical program representations used in text-only reasoning (Gao et al., 2023; Zelikman et al., 2023; Zhang et al., 2024); in contrast, we leverage program representations for understanding concepts in natural images. Notably, DSG performs schema inference with a VLM; compared to fully symbolic inference processes, our framework is capable of leveraging all visual information in the image for interpreting abstractions.

## 3   DEEP SCHEMA GROUNDING

We study the task of reasoning about visual abstractions by concretely focusing on its visual question-answering (VQA) form, although our idea naturally generalizes to other visual interpretation and reasoning tasks. A VQA instance is a tuple of $\langle v, q, a \rangle$, where $v$ is an image, $q$ is a natural language question, and $a$ is the answer to the question. Here, $a$ can take the form of multiple choices or free-form responses in natural language. We assume that each image $v$ has an underlying abstract concept associated with it (e.g., "maze"), which is not associated with concrete visual features, but a set of lifted rules that define the concept. We also assume that $q$ is a question about the abstract concept, and the concept is revealed to all models in question $q$.

The visual abstraction understanding task posits a series of challenges to off-the-shelf VLMs. These questions are centered around concepts with infinite possible instantiations in the real world. As a

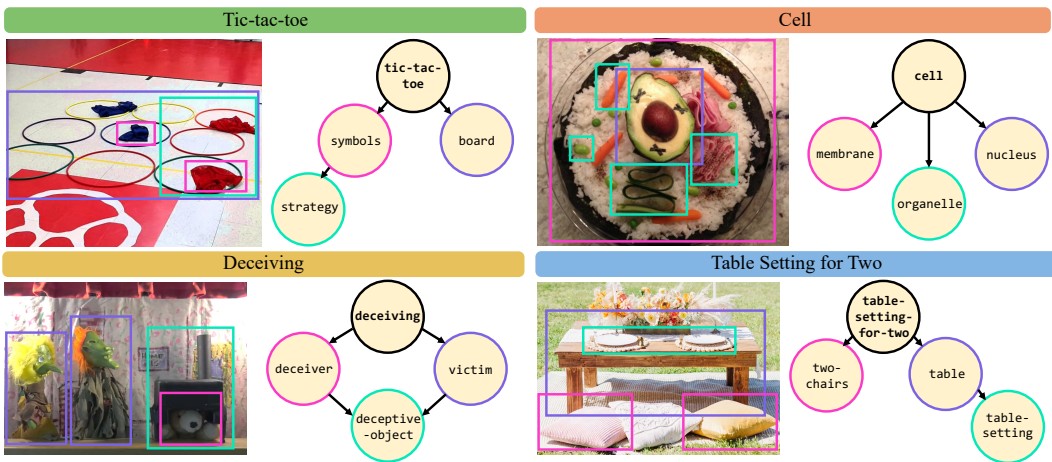

Figure 3: Examples of schemas for concepts and the visual features that they may be grounded to.

result, current VLMs struggle with this problem due to the potential novelty of the scene and the visual abstraction combinations encountered. However, a crucial insight regarding these abstract concepts is the consistency in composition patterns of lower-level concepts, which aggregate to construct the high-level concept; these underlying patterns remain the same across different variations.

Based on these insights, we propose Deep Schema Grounding (DSG), a framework that leverages *schemas* to interpret and reason about visual abstractions. DSG can be built on top of most pretrained vision-language models (VLMs) to perform the visual question-answering task. Illustrated in Figure 2, the DSG framework consists of three main steps: (1) extracting a schema of the concept, (2) hierarchically grounding the schema to the visual input, and (3) leveraging the resolved schema as input to the base VLM. We describe the schema definition and each stage in detail below.

## 3.1 Visual Abstraction Schema

A visual abstraction schema is a concise program that defines a directed acyclic graphical (DAG) representation of a particular concept. As illustrated in Figure 2, each node in the schema corresponds to a subcomponent concept of the higher-level abstract concept. For example, the formation of a maze can be decomposed into three components: the layout, the construction of the walls, and the positioning of the entry and exit of the maze. The dependencies among individual components yield a DAG configuration—in this case, the placement of the entry and the exit of the maze depends on the layout of the maze. From an inference perspective, the dependency graph describes the order in which the model should interpret each component, and how the interpretation of a particular component should be conditioned on the interpretation of one or more previously interpreted components.

The most important feature of a visual abstraction schema is that the definition of such concepts is *universal*, in the sense that the program describes a universal characterization of mazes made of various components, rather than being restricted to specific visual instantiations (e.g., mazes formed by drawn lines). Hence, schemas enable machines to better generalize to novel scenes.

## 3.2 Extracting Schemas of Abstract Concepts

DSG utilizes large language models (LLMs) as a repository of knowledge from which schemas can be acquired. Given that LLMs are trained on extensive corpora of human language data, we hypothesize that they contain human-aligned schemas for a wide range of abstract concepts. Compared to prior works that use LLMs to translate natural language utterances into programs, DSG uses LLMs to extract explicitly structured knowledge of abstract concepts based on the concept's name.

In particular, our goal is to extract a universal schema for abstract concepts that is not instance-specific, but is applicable across diverse visual stimuli. Notably, our prompt to the LLMs is generic and simple, with only one example of a schema for the abstract concept "academia" (which consists of key components: faculty, students, etc.), and a short instruction of the schema generation task. We find that LLMs demonstrate a remarkable ability to extrapolate across a wide array of concepts, and

produce faithful decompositions of the visual abstractions. We detail the prompt and LLM-generated schemas for each concept in Appendix C.

### 3.3 HIERARCHICALLY GROUNDING SCHEMAS ON IMAGES

Instead of directly answering questions given the image and the question inputs, we first hierarchically *ground* individual components in the schema to visual entities in the image. In the maze example in Figure 2, this corresponds to inferring the materials of the walls, the layout of the maze, etc.

Formally, given a schema definition of the concept, DSG hierarchically grounds each component in the concept DAG. The outcome of this process is represented as short text descriptions, such as "wall: coffee beans". We implement this procedure by leveraging pretrained VLMs to predict the most likely grounding of a component given the image and a text query (see Figure 4). Since our grounding procedure is hierarchical, for components that are conditioned on other concepts (e.g., the interpretation of entry-exit is conditioned on the layout), the VLM also takes into account the grounding results of these prerequisite components. This ensures that each step of the grounding process is informed by the previously established context, facilitating more coherent and accurate grounding.

```
gen(layout | concept=maze)
```
Q: "Imagine that the image represents a maze, what is the layout?"
A: Rectangular.

```
gen(walls | concept=maze)
```
Q: "Imagine that the image represents a maze, what are the walls?"
A: Coffee beans.

```
gen(entry-exit | concept=maze, layout)
```
Q: "Imagine that the image represents a maze, and the layout is rectangular, what is the entry and exit?"
A: Coffee cups.

Figure 4: The schema grounding process.

Grounding components following the hierarchical schema structure is crucial for abstract concepts with many possible instantiations. For example, for the "maze" concept, DSG first resolves components that are more coarse and concrete (e.g., layout), then uses those components as conditions to resolve more difficult grounding problems of fine-grained and abstract components (e.g., entry-exit). Figure 3 shows examples of schemas and how they may be grounded onto images. More examples and failure modes can be found in Appendix A.

### 3.4 VISUAL QUESTION-ANSWERING AUGMENTED WITH GROUNDED SCHEMAS

Finally, DSG leverages the grounding of individual components in the image to answer questions. For example, given the concept "maze" and the grounded mapping of individual components: {layout: rectangular, walls: coffee beans, entry-exit: coffee cups}, we augment the final question-answering step by providing the VLM with the component mapping, as well as the full conversation history of the grounding process. In particular, the groundings of the components are included in a text prompt such as: "Imagine that the image represents a maze, and the layout is rectangular, and the walls are coffee beans, and the entry and exit are coffee cups." Compared with baselines that only have access to the abstract concept itself (e.g., "Imagine that the image represents a maze."), our concept grounding system shows significant performance improvement with the resolved schema as holistic context for the image.

Notably, our DSG framework is *inference-only*: it leverages the strong generalization capabilities of LLMs (for proposing schemas) and VLMs (for grounding schemas and reasoning), to better understand visual abstractions. Programmatic schemas are used as the intermediate representation to bridge the explicit commonsense knowledge in LLMs and the visual reasoning capability of VLMs.

## 4 VISUAL ABSTRACTIONS BENCHMARK

In order to evaluate models on visual abstraction reasoning, we propose a new benchmark, the *Visual Abstractions Benchmark* (VAB), illustrated with examples in Figure 5. It consists of 180 images and 3 questions per image, with a total of 540 test examples. VAB is comprised of 12 different abstract concepts spanning 4 categories, where each concept is associated with 15 examples of real-world images. The questions bridge multiple question types: binary-choice, counting, and open-ended questions. We detail the set of abstract concepts and the image and text components of VAB below.

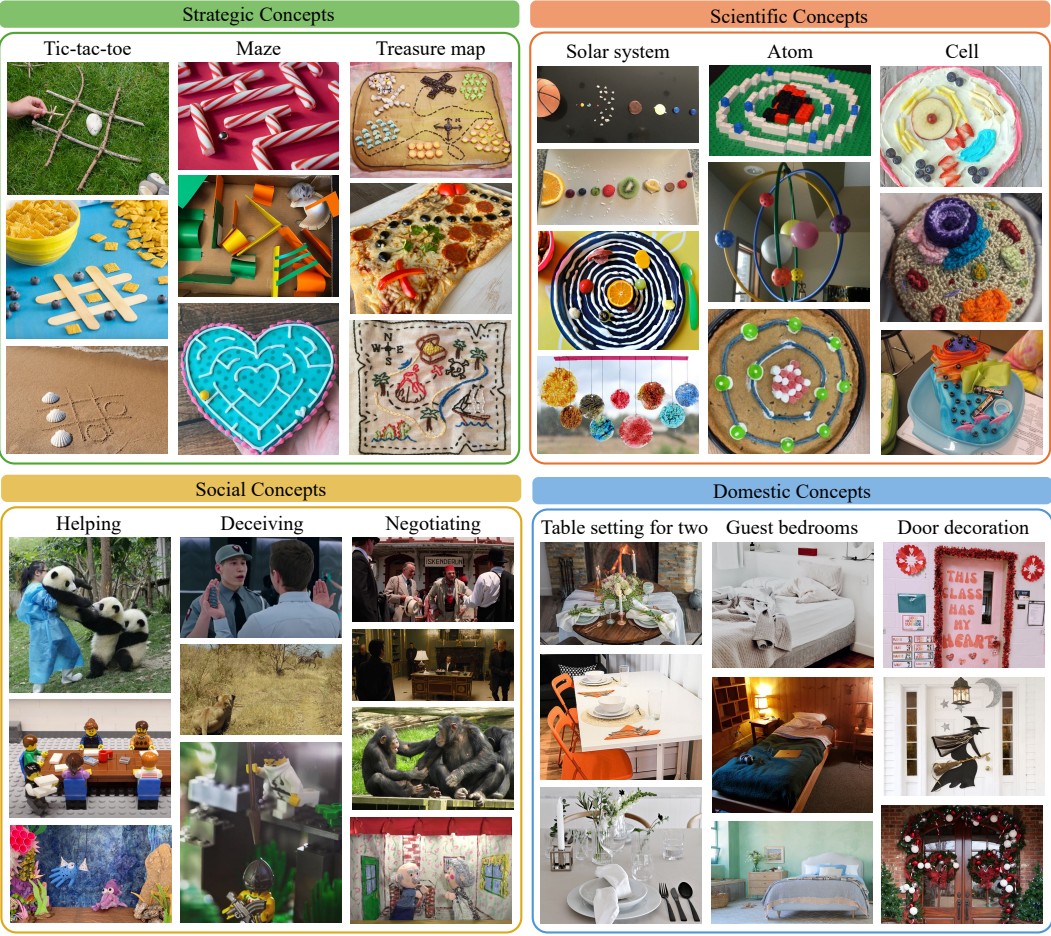

Figure 5: The Visual Abstractions Benchmark comprises diverse, real-world images that represent 12 different abstract concepts across 4 categories.

**Abstract concepts.** We choose 12 different visual abstractions as concepts. All of them can be grounded on different objects and configurations of objects in images. We focus on concepts that are not tied to distinct visual features (e.g., unlike canonical colors and shapes) but instead require higher-level, relational, and lifted patterns. These visual abstractions cover a wide range of visual scenarios and possible linguistic queries, and they can be grouped into four broader categories.

The first category consists of *strategic concepts*, which are games characterized by rules and patterns, including "tic-tac-toe", "maze", and "treasure map". The exact object instantiations and visual features may vary, but humans can easily identify the state of the game when given the concept. The second category consists of *scientific concepts* with no possible visualization in the physical world, including "solar system", "atom", and "cell". While we cannot directly view these concepts, we can create analogies of them based on everyday objects. The third category consists of *social concepts*, which are intentional actions that involve the deduction of roles and relations, including "helping", "deceiving", and "negotiating". Examples of these theory-of-mind concepts in the real world can be seen in many videos and movies; for a single image frame, we must make deductions to assign possible roles to entities in the scene. The fourth category consists of *domestic concepts*, inspired by everyday household tasks (Li et al., 2023). These concepts are goals in household environments that cannot be easily defined by specific arrangements of objects, including "table setting for two", "guest bedrooms", and "door decoration". While humans easily recognize when such objectives are completed, we do not have a simple definition for what exact configuration should be generated.

**Images and question-answers.** We curate 15 images for each abstract concept from the Internet. Our criteria for the images are to be real, natural, and captured in the physical world. Our final set of images is diverse in terms of subjects, environments, and domains. See Figure 5 for examples.

Table 1: DSG outperforms prior works across all questions and different question types in the Visual Abstractions Benchmark. Results are averaged over 5 runs.

|  | All | Counting | Binary | Open |
|---|---|---|---|---|
| ViperGPT (Surís et al., 2023) | 0.260 | 0.191 | 0.426 | 0.141 |
| VisProg (Gupta & Kembhavi, 2023) | 0.304 | 0.249 | 0.385 | 0.255 |
| LLaVA (Liu et al., 2024) | 0.415 | 0.344 | 0.552 | 0.321 |
| InstructBLIP (Dai et al., 2024) | 0.437 | 0.322 | 0.538 | 0.392 |
| GPT-4o (OpenAI, 2024) | 0.664 | 0.604 | 0.655 | 0.693 |
| DSG (ours) | **0.730** | **0.704** | **0.690** | **0.776** |
| Human | 0.846 | 0.816 | 0.833 | 0.868 |

Table 2: DSG improves the performance of GPT-4o across categories of abstract concepts.

|  | Strategic | Scientific | Social | Domestic |
|---|---|---|---|---|
| GPT-4o (OpenAI, 2024) | 0.653 $\pm$0.052 | 0.646 $\pm$0.074 | 0.647 $\pm$0.163 | 0.708 $\pm$0.082 |
| DSG (ours) | **0.751** $\pm$0.088 | **0.693** $\pm$0.126 | **0.723** $\pm$0.027 | **0.754** $\pm$0.104 |

We create 3 types of natural language questions for each abstract concept in VAB. These questions ask about different aspects of the images and take various forms, including binary-choice questions, counting questions, and general open-ended questions. In total, there are 6 types of counting questions, 14 types of binary-choice questions, and 16 types of open-ended questions. An example of a binary-choice question is "Imagine that the image represents a table setting for two. What is the configuration of the seats? Describe the answer as across or adjacent."; an example of a counting question is "Imagine that the image represents a cell. Excluding the nucleus, how many types of organelles are there?"; an example of an open-ended question is "Imagine that the image represents a treasure map. Based on the image, find the starting location; what is the main color of the map region that is physically closest to the starting location?" The full Visual Abstractions Benchmark comprises 540 of such questions.

Answers to all image-question pairs are labeled by 5 human annotators from Prolific; we take their modal response as the answer, to be used for exact-match text accuracy. While the Visual Abstractions Benchmark focuses on multiple-choice questions and answers, we also provide a free-response variant of VAB. We present examples for each type of question along with corresponding images and answers in Appendix B, and release our benchmark here.

## 5 RESULTS

We validate Deep Schema Grounding in comparison to baseline VLMs and integrated LLMs with APIs on the Visual Abstractions Benchmark. We first compare our work with ViperGPT (Surís et al., 2023), VisProg (Gupta & Kembhavi, 2023), LLaVA (Liu et al., 2024), InstructBLIP (Dai et al., 2024), and GPT-4o (OpenAI, 2024). Then, we ablate different components of the DSG framework. Finally, we show that DSG is model-agnostic and can benefit both closed-source and open-source models. Baseline models are given the abstract concept itself in the prompt along with the question (e.g., "Imagine that the image represents a maze."), but without DSG's resolved schema of the concept. All results are averaged over 5 runs. Here, we systematically evaluate multiple-choice answers, where accuracy is calculated as exact-match accuracy. In Appendix A, we provide additional comparisons with a graded accuracy metric based on the degree of alignment to human judgments, and results on free-form answers using BERTScores (Zhang et al., 2020) and LLMs (Kamalloo et al., 2023).

**VLMs benefit from decomposing abstract concepts into primitive-level components.** Table 1 summarizes the multiple-choice accuracy of DSG with GPT-4o as the base VLM, in comparison to that of prior works across 5 runs. DSG significantly outperforms VLMs and integrated LLMs with APIs across all questions and all types of questions. In particular, our method demonstrates a boost in accuracy of 6.6 percent points overall ($\uparrow$ 9.9% relative improvement) compared to GPT-4o. Across different question types, DSG shows the largest performance improvement of 10 percent points ($\uparrow$ 16.6% relative improvement) for counting-based questions, such as "How many entities are

Table 3: Summarization of DSG ablations; the full DSG framework shows strongest performance.

| | schema | grounding | hierarchy | context | All | Counting | Binary | Open |
|---|---|---|---|---|---|---|---|---|
| + schema | ✓ | ✗ | ✗ | ✗ | 0.681 | 0.629 | 0.672 | 0.709 |
| + grounding | ✓ | ✓ | ✗ | ✗ | 0.693 | 0.727 | 0.660 | 0.710 |
| + hierarchy | ✓ | ✓ | ✓ | ✗ | 0.701 | 0.713 | 0.671 | 0.723 |
| DSG (ours) | ✓ | ✓ | ✓ | ✓ | **0.730** | **0.704** | **0.690** | **0.776** |

helping?" We hypothesize that the significant improvement in accuracy for these questions is due to more accurate labels for objects with DSG, as the schema grounding step will concretely associate concepts such as "the entity who is helping" with particular parties in the image (e.g., pandas). These questions usually require a more sophisticated reasoning process to answer correctly. In comparison, for binary-choice questions, there may exist more superficial correlations that off-the-shelf VLMs can rely on. Importantly, we also report a random human annotator's accuracy of adhering to the modal human response to highlight the difficulty of problems in VAB.

In Table 2, we more closely analyze the performance improvement of DSG on GPT-4o across concept categories. We report the average performance and the standard deviation for 5 runs. We see that DSG outperforms GPT-4o across the four categories. In particular, our method demonstrates a strong boost in accuracy of 9.8 percent points ($\uparrow 15\%$ relative improvement) for strategic concepts, and 7.6 percent points ($\uparrow 11.7\%$ relative improvement) for social concepts. In Appendix A, we include qualitative examples of cases where DSG succeeds and fails in correctly grounding schemas. Overall, DSG's decomposition-based framework improves performance of VLMs across abstract concepts.

Notably, we also analyze the performance of GPT-4v OpenAI (2023) and DSG with GPT-4v as the base model, compared to that of GPT-4o. We summarize results as follows: GPT-4v yields an overall accuracy of $0.653$, GPT-4o yields $0.664$, DSG with GPT-4v yields $0.707$, and DSG with GPT-4o yields $0.730$. We see that DSG significantly improves the performance of both base models. Importantly, DSG with GPT-4v as the base model outperforms GPT-4o by $4.3$ percent points, and showcases the potential for DSG to improve the visual reasoning performance of weaker models.

**Explicit hierarchical grounding of schemas is better than implicit and sequential grounding.** We present ablation studies of different modules in DSG in Table 3. In particular, we evaluate (1) a variant of DSG that is given only the LLM-generated schema for chain-of-thought reasoning within a single query, without using explicit grounding steps (+ schema), (2) a variant where DSG performs decomposed grounding sequentially instead of hierarchically (+ grounding), and (3) a hierarchical variant (+ hierarchical) where in the final question-answering step, we provide the model with only the grounding result of each schema component, instead of a full history of its grounding process.

With the first variant, we test whether GPT-4o can use the schema appropriately without a structured thinking process. We explore the second variant to test whether decomposing the visual abstraction into a set of key elements is sufficient for improving the understanding of abstract concepts, and evaluate the role of hierarchical dependencies in lower-level concepts for grounding. Finally, we present results on the third variant to test whether the full grounding trace of DSG is helpful during the final question-answering step. The results show that all these design choices help: a schema only does not lend itself to much improved performance, and hierarchical grounding from concrete to abstract symbols, as well as the inclusion of the full context, improves performance in many concepts.

**DSG improves other (open-source) VLMs, too.** Tables 4 and 5 summarize the results of DSG with open-source LLaVA as the base VLM. DSG yields consistent improvement overall ($\uparrow 7.0\%$ relative improvement). However, in contrast to GPT-4o, both LLaVA and DSG yield poor results on counting questions, as the base VLM struggles at enumeration regardless of its abstraction understanding.

Table 4: DSG improves the performance of the open-source LLaVA model.

| | All | Counting | Binary | Open |
|---|---|---|---|---|
| LLaVA (Liu et al., 2024) | 0.415 | **0.344** | 0.552 | 0.321 |
| DSG (ours) | **0.444** | **0.344** | **0.590** | **0.354** |

Table 5: Comparison of LLaVA and DSG across categories; DSG outperforms the base VLM.

|  | Strategic | Scientific | Social | Domestic |
|---|---|---|---|---|
| LLaVA (Liu et al., 2024) | 0.407 ±0.038 | 0.333 ±0.101 | 0.378 ±0.126 | 0.541 ±0.091 |
| DSG (ours) | **0.430** ±0.046 | **0.378** ±0.119 | **0.393** ±0.154 | **0.578** ±0.096 |

## 6 DISCUSSION

Our framework demonstrates that by explicitly grounding conceptual schemas through large pretrained vision and language models, we can achieve better visual abstraction understanding in diverse real-world instantiations. We find that LLMs generally possess well-structured knowledge of concepts, outputting comprehensive schemas that are accurately decomposed into more concrete symbols and their dependencies. The extracted schemas effectively capture the concise essence of a concept, such as the answer to our initial question of *what makes a maze look like a maze*. However, as DSG relies on pre-trained LLMs to generate schemas and does not restrict the schemas to a set of specified symbols, it is possible that the schemas may contain harmful biases based on the concept given, or may include symbols that are difficult for VLMs to interpret.

In particular, we observe that current VLMs struggle to ground challenging schema components involving spatial constraints. DSG does not explicitly improve this capability or specify how to parameterize spatial configurations. While DSG can generate schema components that contain spatial relationships, and try to ground them by describing them in language, it remains limited by the VLMs' abilities to conduct such spatial understanding. For example, let us consider a case where the extracted schema for the concept maze includes a symbol representing "an unobstructed path from entry to exit". That such a path exists in the maze—typically only one and often one that is difficult to find—is a defining feature of mazes, yet is challenging for VLMs to correctly identify. VLMs' lack of spatial understanding weakens DSG's ability to interpret complex and spatially grounded components. Scaling DSG to better understand concepts that require precise layouts of schema components is difficult but crucial for future work.

A key aspect of DSG is that our framework does not limit the expressiveness of schemas to a specific set of symbols or the grounding process to a specific set of answers. The only prior we inject into DSG is the hierarchical process of resolving the schema—the interface between the LLM and VLMs is unconstrained language. In theory, DSG can propose and ground any concept and any individual component, allowing the LLM and VLM to determine what defines a visual abstraction without any restrictions. But such flexibility may also render the schema ambiguous and not universal enough to improve downstream tasks. However, we see that the empirical evidence in our paper supports DSG, and we believe that DSG serves as a proof of concept for the kind of inductive biases that should be incorporated into visual reasoning systems—a structured thinking process guided by flexible but explicit knowledge can augment visual abstraction interpretation.

While we have made progress on understanding what makes a maze look like a maze, in proposing DSG as a framework for leveraging explicit schemas backed by powerful LLMs and VLMs, we are far from solving all reasoning across abstract concepts as humans do. There may be ways to improve the grounding of individual components, as well as more efficient priors that can be incorporated into the system. We leave such exploration to future work.

## 7 CONCLUSION

We propose Deep Schema Grounding as a promising approach to understanding visual abstractions. DSG leverages *schemas* of concepts to decompose abstract concepts into subcomponents and model their dependencies. Our framework extracts explicitly structured schemas from large language models and hierarchically grounds them to images with vision-language models. On the Visual Abstractions Benchmark, a visual question-answering benchmark composed of real-world images with diverse underlying abstract concepts, DSG demonstrates significant improvements compared to base VLMs. DSG is a step toward interpreting visual abstractions as humans do; more remains to be done in the challenge of visual abstraction reasoning.

ACKNOWLEDGMENTS

We thank Michael Li and Weiyu Liu for providing valuable feedback. This work is in part supported by the Stanford Institute for Human-Centered Artificial Intelligence (HAI), ONR N00014-23-1-2355, ONR YIP N00014-24-1-2117, NSF RI #2211258, Air Force Office of Scientific Research (AFOSR) YIP FA9550-23-1-0127, the Center for Brain, Minds, and Machines (CBMM), the MIT Quest for Intelligence, MIT–IBM Watson AI Lab, and Analog Devices. JH is also supported by the Knight Hennessy Scholarship and the NSF Graduate Research Fellowship.

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

# SUPPLEMENTARY FOR:
# WHAT MAKES A MAZE LOOK LIKE A MAZE?

The appendix is organized as the following. In Appendix A, we present comparisons of Deep Schema Grounding to prior works with a graded accuracy metric, comparisons on free response generation, results on single- *v.s.* multi-round reasoning, results with language-only baselines, ablations on schema complexity, qualitative examples of DSG, and examples of hierarchical grounding. In Appendix B, we show more examples of the Visual Abstractions Benchmark, explain our benchmark construction procedure, describe our annotation collection process, and release our dataset. In Appendix C, we provide our prompt to the LLM and all extracted schemas, as well as results from a human study evaluating the quality of the generated schemas.

## A  ADDITIONAL RESULTS

### A.1  GRADED ACCURACY COMPARISON

Due to variation in human responses on the Visual Abstractions Benchmark—arising from natural errors in annotation or potential ambiguity in questions—we compare Deep Schema Grounding to prior works with a *graded* accuracy metric, based on degree of alignment to human judgements. Predicted answers each receive $\frac{K}{5}$ points, where $K$ is the number of human annotators out of 5 that agree with the prediction. In Table 6, we report results with the graded accuracy metric over 5 runs. DSG consistently outperforms previous works, showing an overall accuracy increase of 6.2 percent points ($\uparrow 10\%$ relative improvement), and an accuracy boost in counting questions of 10.8 percent points ($\uparrow 20.8\%$ relative improvement). We observe a similar trend in performance as with modal response accuracy reported in the main text—more improvement is seen in counting questions.

Table 6: Comparison of DSG to prior works with a graded accuracy metric based on degree of alignment to human judgements.

|  | All | Counting | Binary | Open |
|---|---|---|---|---|
| ViperGPT (Surís et al., 2023) | 0.254 | 0.200 | 0.413 | 0.136 |
| VisProg (Gupta & Kembhavi, 2023) | 0.293 | 0.227 | 0.380 | 0.242 |
| LLaVA (Liu et al., 2024) | 0.400 | 0.327 | 0.548 | 0.298 |
| InstructBLIP (Dai et al., 2024) | 0.420 | 0.307 | 0.528 | 0.368 |
| GPT-4o (OpenAI, 2024) | 0.622 | 0.519 | 0.655 | 0.632 |
| DSG (ours) | **0.684** | **0.627** | **0.681** | **0.708** |
| Human | 0.789 | 0.728 | 0.784 | 0.816 |

### A.2  COMPARISON ON FREE RESPONSE GENERATION

We present comparisons between DSG and GPT-4o on free response generation in Table 7, with accuracy first evaluated by BERTScores Zhang et al. (2020). We see that DSG outperforms GPT-4o across all questions and question types.

Table 7: Performance on free response generation; DSG outperforms the base VLM.

|  | All | Counting | Binary | Open |
|---|---|---|---|---|
| GPT-4o (OpenAI, 2024) | 0.900 | 0.917 | 0.942 | 0.858 |
| DSG (ours) | **0.908** | **0.949** | **0.946** | **0.859** |

Additionally, we report free-response results by incorporating an LLM-based strategy for evaluating free-response answers (Kamalloo et al., 2023). In particular, we use zero-shot evaluation via LLM prompting, which includes the ground truth and predicted answer in the prompt, to verify the correctness of the prediction. We use GPT-4 as the base LLM, and compare GPT-4o with DSG. In Table 8, we see that DSG outperforms GPT-4o overall, with particularly strong results on counting and open-ended questions, and shows stronger signal than with BERTScores. However, performance on binary questions is lower, which we hypothesize is due to some of the components being potentially misleading or ambiguous in binary contexts.

Table 8: Performance on free response generation with Kamalloo et al. (2023)'s GPT-4-based metric for evaluating open-domain QA models.

|  | All | Counting | Binary | Open |
|---|---|---|---|---|
| GPT-4o | 36.67 | 45.56 | **55.24** | 14.58 |
| DSG (ours) | **37.78** | **53.33** | 49.52 | **21.25** |

## A.3 RESULTS ON SINGLE-ROUND VS. MULTI-ROUND REASONING

The key idea behind DSG lies in schema decomposition (e.g., in the form of extracting components: layout, walls, and entry and exit) to guide the VLM to interpret the scene. DSG introduces an explicit guidance to the VLM system, by first generating a decomposition of the target concept, parsing the image, then answering the question. The specific method of processing these questions—whether in a single query or across multiple queries—is not fundamentally important. While in theory, these steps could be merged into a single CoT-style prompt (Wei et al., 2022), our following experiments show that multi-round execution with explicit grounding steps significantly improves performance in practice, at least for the current vision-language models.

Here, we compare results of (1) GPT-4o, (2) a single-round CoT baseline given schema components (e.g. the VLM is given components for reasoning in a single query, without using the explicit grounding steps), (3) a single-round CoT baseline given the full hierarchical schema (also in Table 3), and (4) our full multi-round DSG framework. All results are averaged over 5 runs. In Table 9, we see that DSG's multi-round execution outperforms single-round, CoT-style baselines, likely due to its explicit guidance and grounding.

Table 9: Comparison of DSG with single- and multi-round grounding of the schema, with results averaged over 5 runs.

|  | All | Counting | Binary | Open |
|---|---|---|---|---|
| GPT-4o | 0.664 | 0.604 | 0.655 | 0.693 |
| Single-round with schema components | 0.667 | 0.607 | 0.661 | 0.696 |
| Single-round with hierarchical schema | 0.681 | 0.629 | 0.672 | 0.709 |
| Multi-round with hierarchical schema (DSG) | **0.730** | **0.704** | **0.690** | **0.776** |

## A.4 LANGUAGE-ONLY BASELINES

We additionally report results on language baselines. The first is a language-only baseline, where GPT-4o only has access to the question itself without input image or schema. The second is a language-with-schema baseline, where GPT-4o has access to the question and the DSG grounded schema (the schema is extracted from the image and provides context), but the final query itself does not have access to the image. We report results over 5 runs, and present results in Table 10.

Table 10: Comparison with language-only baseline and baseline with DSG's grounded schema, with results averaged over 5 runs.

|  | All | Counting | Binary | Open |
|---|---|---|---|---|
| Language-only | 0.397 | 0.300 | 0.510 | 0.335 |
| Language w/ DSG schema | **0.522** | **0.440** | **0.530** | **0.546** |

Notably, the language-only baseline performs worse than all open-source and closed-source VLMs, showing that the visual input is essential for achieving higher accuracy on the benchmark. We note that the language-only baseline outperforms integrated LLMs with API baselines, as these methods tend to fail in execution. With DSG's grounded schema, the language model performs significantly better, highlighting the importance of the holistic context provided by the grounded schema. We see that in open-ended questions, the grounded schema significantly improves performance of the language-only baseline. We hypothesize that this is because the schema helps rule out implausible answer choices by leveraging the grounded components.

A.5 SCHEMA COMPLEXITY ABLATION

We highlight that there's a tradeoff between encouraging more detailed components in a schema and avoiding both components that are less broadly applicable across all image instantiations as well as potential failures of current VLMs in handling such detailed concepts, especially more spatially-focused ones. As capabilities of VLMs increase, we believe DSG's performance will scale accordingly, and we can increase the detail of the schema without making this tradeoff. In addition, the complexity of the generated schema is currently influenced by the number of components in the example prompt. As LLMs become less dependent on the form of the initial prompt, they can decide the schema without being influenced by the bias in the example.

To explore the impact of schema complexity and better understand the capabilities of current VLMs, we report ablations results varying the numbers of components in the example prompt: 3, 5, 7, and 9. The example prompts are based on the same abstract concept `academia` expanded to different levels of detail.

```
With 3 components:
gen(concept=academia) =
   gen(faculty | concept=academia)
   gen(students | concept=academia)
   gen(research-output | concept=academia, faculty, students)
```

```
With 5 components:
gen(concept=academia) =
   gen(faculty | concept=academia)
   gen(students | concept=academia)
   gen(courses | concept=academia, faculty, students)
   gen(research-output | concept=academia, faculty, students, courses)
   gen(funding | concept=academia, research-output)
```

```
With 7 components:
gen(concept=academia) =
   gen(faculty | concept=academia)
   gen(students | concept=academia)
   gen(departments | concept=academia, faculty, students)
   gen(courses | concept=academia, departments, faculty, students)
   gen(research-output | concept=academia, departments, faculty, students)
   gen(funding | concept=academia, research-output, departments)
   gen(infrastructure | concept=academia, funding, departments, courses)
```

```
With 9 components:
gen(concept=academia) =
   gen(faculty | concept=academia)
   gen(students | concept=academia)
   gen(departments | concept=academia, faculty, students)
   gen(courses | concept=academia, departments, faculty, students)
   gen(research-output | concept=academia, departments, faculty, students)
   gen(funding | concept=academia, research-output, departments)
   gen(infrastructure | concept=academia, funding, departments, courses)
   gen(administration | concept=academia, infrastructure, departments, faculty, students)
   gen(community-engagement | concept=academia, research-output, administration, students,
       faculty)
```

Table 11: Ablation with varying number of components in the example prompt.

|                | All       | Counting  | Binary    | Open      |
|----------------|-----------|-----------|-----------|-----------|
| DSG w/ 3 comp. | **0.730** | **0.689** | **0.700** | **0.771** |
| DSG w/ 5 comp. | 0.691     | 0.622     | 0.671     | 0.733     |
| DSG w/ 7 comp. | 0.685     | 0.667     | 0.662     | 0.713     |
| DSG w/ 9 comp. | 0.676     | 0.700     | 0.657     | 0.683     |

In Table 11, the results demonstrate that accuracy decreases as the number of components in the example prompt increases, likely due to current VLMs struggling with grounding more specific components. We note that all DSG variants still outperform the GPT-4o baseline. Interestingly, our findings also highlight a tendency for LLMs to generate shorter schemas, even when exposed to more detailed example prompts. In Table 12 we show the number components in the prompts and the mean number of components in the resulting schemas, averaged over all abstract concepts.

Table 12: Number of components in schema and the resulting average number of components in the generated schemas.

|  | Generated comp. # |
|---|---|
| 3 comp. | 3.00 |
| 5 comp. | 4.66 |
| 7 comp. | 5.75 |
| 9 comp. | 7.00 |

Notably, although the number of components in the example prompt does affect the average number of components in resulting schemas, this effect is less prominent and the resulting average plateaus. This phenomenon indicates that schemas may potentially converge to shorter lengths. And while current VLMs' limitations constrain schema complexity, as these capabilities improve with development, we expect DSG's performance to scale, enabling our framework to adopt more detailed schemas without sacrificing effectiveness.

## A.6 QUALITATIVE EXAMPLES

We present examples of DSG's resolved schemas in Figure 6; in particular, we highlight cases where DSG's schema grounding process errors in red.

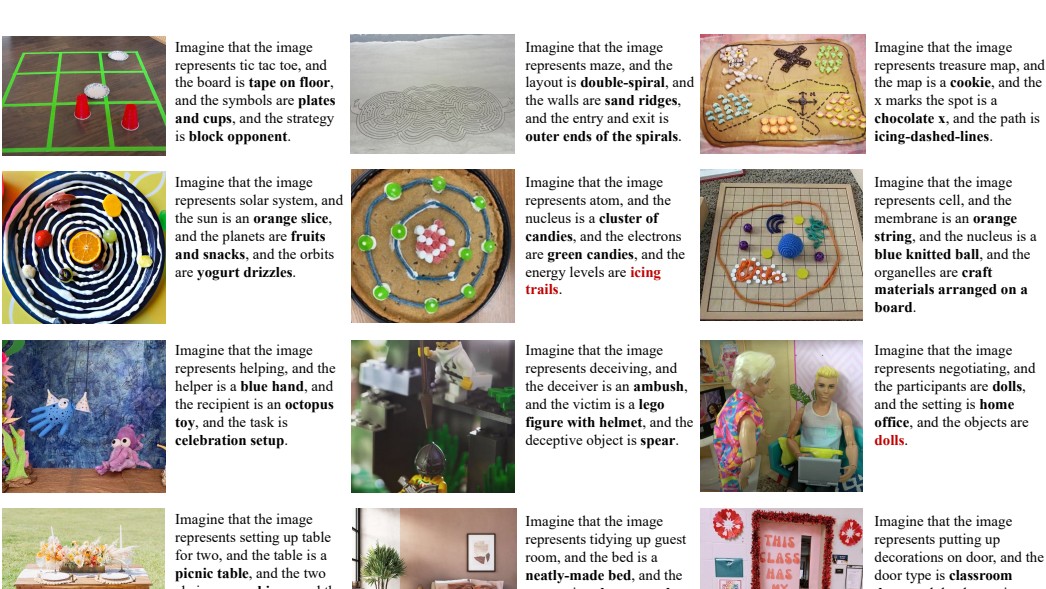

Figure 6: Examples of DSG's resolved schemas across different abstract concepts; in red we highlight failure cases where inference of individual components errors.

## A.7 HIERARCHICAL GROUNDING EXAMPLE

In Figure 7, we highlight an example of how DSG's hierarchical inference improves schema grounding accuracy compared to a sequential version of DSG.

## A.8 COMPUTING RESOURCES

We use OpenAI's API for GPT-4o (OpenAI, 2024), ViperGPT (Surís et al., 2023), and VisProg (Gupta & Kembhavi, 2023). Open-sourced models, LLaVA (Liu et al., 2024) and InstructBLIP (Dai et al.,

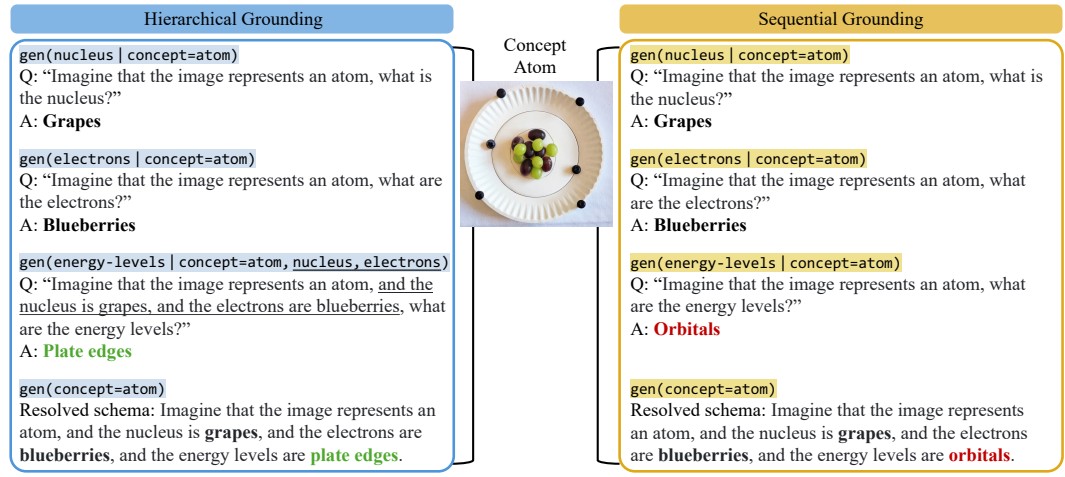

Figure 7: In comparison to DSG, which hierarchically grounds each component, a sequential variant errors when grounding a more fine-grained symbol, `energy-levels`, without conditioning on prior predictions of more concrete symbols, `nucleus` and `electrons`.

2024), and the API calls of the aforementioned integrated LLMs with APIs, were run inference-only with 1 A40 on an internal cluster.

# B    VISUAL ABSTRACTIONS BENCHMARK

## B.1    BENCHMARK EXAMPLES

In Figure 9, we provide visual question-answering examples of VAB across concept types and question types.

## B.2    BENCHMARK CONSTRUCTION

We constructed the Visual Abstractions Benchmark inspired by areas where abstract concepts would occur in the real-world, including strategic games which can be played across instantiations, scientific concepts where many classroom science projects exist, social concepts with theory-of-mind examples, and domestic concepts with tasks that do not have predefined and exact configurations of end states. Images were manually curated from the Internet, primarily using Google Image Search. We searched for diverse real-world representations of each concept (e.g., tic-tac-toe on the beach). Images were chosen such that each image is (a) faithful to the given concept, (b) diverse compared to other chosen images, (c) real-world, and (d) high-quality; this process is difficult to procedurally replicate. We then wrote general questions for each concept, aiming for broad applicability and alignment with the images. These questions focus on reasoning grounded in the abstract concept rather than specific visual details of individual images. Once images and questions were curated, they were sent to human annotators via Prolific for labeling.

## B.3    ANNOTATION

We collect annotations for the Visual Abstractions Benchmark through Prolific; 5 human annotators labeled each question-image pair. Figure 8 shows our annotation interface, built from jsPsych. Each annotator was compensated at the rate of $12 USD per hour, and no potential risk was incurred by the annotators.

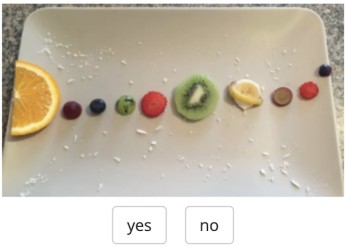

yes    no

Imagine that the image represents solar system. Is there an item that represents the sun?

Figure 8: Example of our annotation interface shown to Prolific annotators.

## B.4    RELEASE

We release the Visual Abstractions Benchmark here. We group all images by their corresponding abstract concepts. Each image is carefully curated and manually audited to be safe for release.

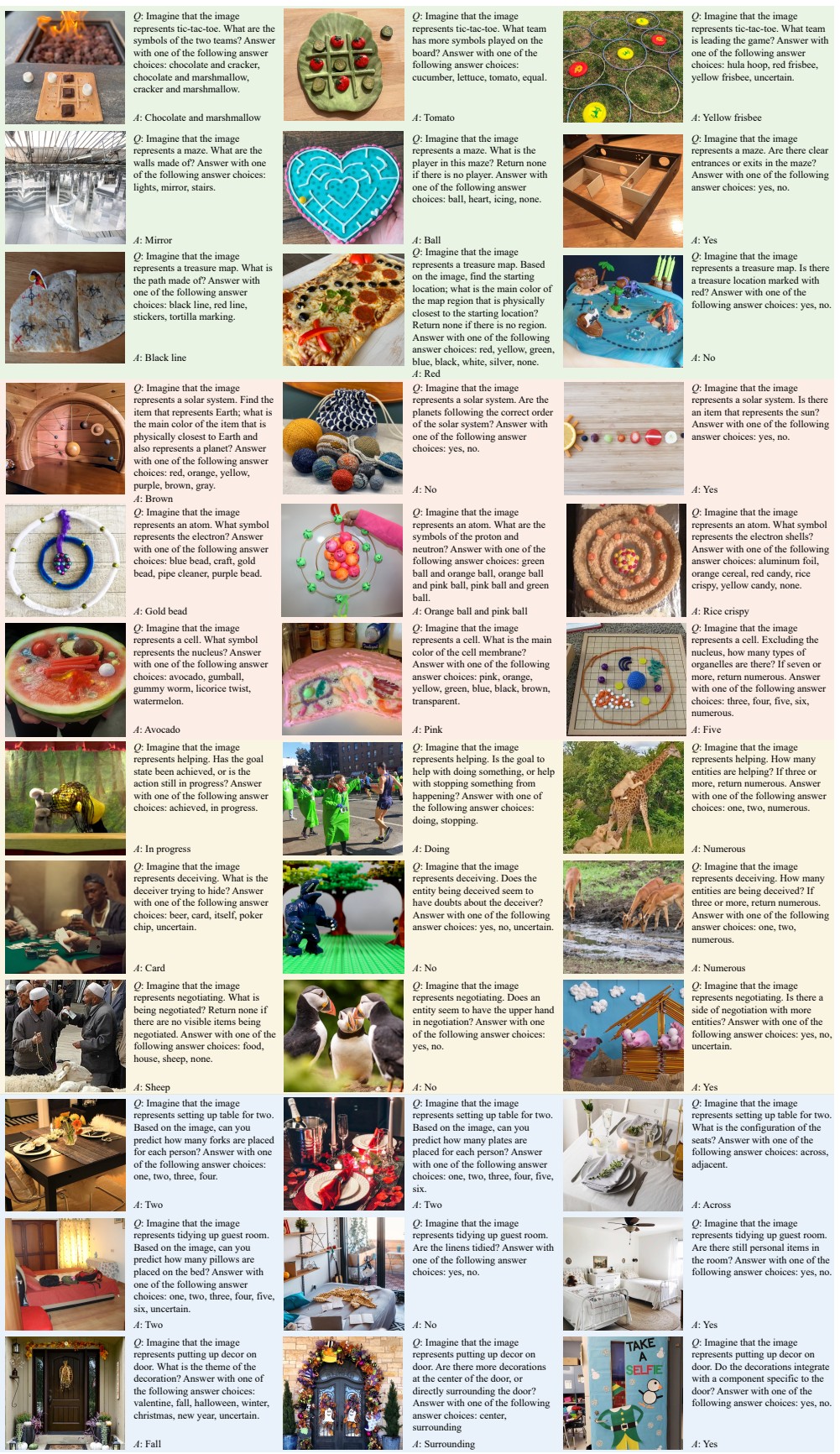

Figure 9: Examples of visual question-answering pairs in the Visual Abstractions Benchmark.

## C  LLM-GENERATED SCHEMAS

### C.1  PROMPT

Below, we provide the prompt used as input to GPT-4 to extract schemas for visual abstractions; we substitute [`abstract-concept`] for a given concept.

```
Can you give me a program representing the schema for a concept? For example,
gen(concept=academia) =
    gen(faculty | concept=academia)
    gen(students | concept=academia)
    gen(research-output | concept=academia, faculty, students)
Please do the same for gen(concept=[abstract-concept]) in the same format without explanation.
      Keep the program simple with four or less components. Use only the most necessary parts
    of the schema that can be mapped to objects in an image.
```

### C.2  ALL SCHEMAS

Here, we present all LLM-generated schemas for the Visual Abstractions Benchmark.

*Strategic concepts.*

```
gen(concept=tic-tac-toe) =
    gen(board | concept=tic-tac-toe)
    gen(symbols | concept=tic-tac-toe)
    gen(strategy | concept=tic-tac-toe, symbols)

gen(concept=maze) =
    gen(layout | concept=maze)
    gen(walls | concept=maze)
    gen(entry-exit | concept=maze, layout)

gen(concept=treasure-map) =
    gen(map | concept=treasure-map)
    gen(x-marks-the-spot | concept=treasure-map)
    gen(path | concept=treasure-map, map)
```

*Scientific concepts.*

```
gen(concept=solar-system) =
    gen(sun | concept=solar-system)
    gen(planets | concept=solar-system)
    gen(orbits | concept=solar-system, sun, planets)

gen(concept=atom) =
    gen(nucleus | concept=atom)
    gen(electrons | concept=atom)
    gen(energy-levels | concept=atom, nucleus, electrons)

gen(concept=cell) =
    gen(membrane | concept=cell)
    gen(nucleus | concept=cell)
    gen(organelles | concept=cell)
```

*Social concepts.*

```
gen(concept=helping) =
    gen(helper | concept=helping)
    gen(recipient | concept=helping)
    gen(task | concept=helping, helper, recipient)

gen(concept=deceiving) =
    gen(deceiver | concept=deceiving)
    gen(victim | concept=deceiving)
    gen(deceptive-object | concept=deceiving, deceiver, victim)

gen(concept=negotiating) =
    gen(participants | concept=negotiating)
    gen(setting | concept=negotiating)
    gen(objects | concept=negotiating, participants)
```

*Domestic concepts.*

```
gen(concept=setting-up-table-for-two) =
   gen(table | concept=setting-up-table-for-two)
   gen(two-chairs | concept=setting-up-table-for-two)
   gen(table-setting | concept=setting-up-table-for-two, table)

gen(concept=tidying-up-guest-room) =
   gen(bed | concept=tidying-up-guest-room)
   gen(storage | concept=tidying-up-guest-room)
   gen(cleanliness | concept=tidying-up-guest-room, bed, storage)

gen(concept=putting-up-decorations-on-door) =
   gen(decoration-type | concept=putting-up-decorations-on-door)
   gen(door-type | concept=putting-up-decorations-on-door)
   gen(tools | concept=putting-up-decorations-on-door, decoration-type)
```

## C.3 HUMAN STUDY OF SCHEMA QUALITY

To evaluate the correctness and quality of the generated schemas, we conduct a human study using Prolific, with 20 participants. Each participant was asked 3 questions about each schema. The questions are as follow for an example concept `maze`:

```
Here you are given a concept, maze, and the core components underlying it: layout, walls,
   entry-exit. How well do these core components represent this concept on a scale of 1-7?
Here you are given a concept, maze, and the core components underlying it: layout, walls,
   entry-exit. How many components out of the ones listed accurately represent part of this
   concept?
Here you are given a concept, maze, and the core components underlying it: layout, walls,
   entry-exit. How many more components, if any, would you add to accurately represent this
   concept?
```

We collect 3 responses for each of 12 concepts with 20 responses each, and yield a total of 720 human annotated answers. In Table 13, we see that (a) participants rated schemas as reasonably representative, (b) $86.3\%$ of the components were considered accurate by participants, and (c) participants suggested adding 1.6 components on average per schema. The concept that ranks highest for the first question is the concept `cell` with a rating of 6.15, and lowest as the concept `negotiating` with 4.50. The concept `solar-system` had the highest number of components to keep at $95\%$, while the concept `negotiating` is the lowest again at $61.7\%$. The concept `tic-tac-toe` was the most complete, with an average of 1 concept that participants might add, while the concept `solar-system` had an average of 2.65. Though there is still room for improvement, the schemas generated by the LLM are largely accurate and aligned with human expectations. Importantly, none of the schemas were judged as fully invalid, showing the robustness of DSG's approach.

Table 13: Prolific study on quality of generated schemas, with results averaged over 20 participants.

|  | Representativeness (1-7) | # comp. to keep | # comp. to add |
| --- | --- | --- | --- |
| Participant responses | 5.74 | 2.59 | 1.59 |

