# OpenReview forum: "What Makes a Maze Look Like a Maze?"
_ICLR.cc/2025/Conference — ICLR 2025 Poster_

### Official Review · Reviewer_8h5R · 2024-10-22

**Soundness:** 3
**Presentation:** 3
**Contribution:** 3
**Rating:** 8
**Confidence:** 3

**Summary:**

This paper introduces Deep Schema Grounding (DSG), a framework that extracts structured representations from LLMs to grounding and reasoning over visual abstractions. DSG leverages the inner knowledge of LLMs to generate schema descripting the abstraction concept. The content within abstract schema will be later grounding into corresponding compoents in the image by Large Vision Model via a hierarchically process. To evaluate the effectiveness of DSG, a new benchmark, Visual Abstraction Benchmark (VAB), is purposed in paper to evaluate model's understanding in visual abstractions.  VAB consists of 540 test examples consists of 180 images presenting 12 abstraction concepts spanning 4 categoris (15 examples each).

The paper do comprehenisve experiment comparing DSG with 5 current VLMs. The result shows DSG can boost 6.6 percent accuracy oveall compared to GPT-4o while current VLMs shows a big gap between human performance. Further abalation study pinpoint that each modules in DSG is necessary and contribute to its success. DSG can boost model performance on both open-source and close-source model but authors notes the challenge still exist in visual abstraction field.

**Strengths:**

1.  Schema-based framework: The paper start from cognitive science and theory-theory to purpose a schema-based concept representation framework. The process of representing high-level abstract concept and later grounding in actual image bring boost in model's ability of abstract understanding. The pre-generetad abstract concept schema seems helpful on mitigate halluciations (16.6% relative improvement in counting-based questions) and is beneficial for both open and closed source models.
2. Curately designed Visual Abstraction Benchmark (VAB): The author provide detail explanation the reason behind VAR design choice. Detailed description of data collection and curation process is provided. The dataset is released for public use. As abstraction can play important role in understanding concept and ease the learning process, I believe this benchmark is a good step for solving the abstraction problem.
3. A comprehensive experiment: the experiment extensively test current LVMs and point out their weakness in visual abstraction understanding.  Further experiment demonstrate the effectiveness of DSG and the ablation studies prove the design choice of each module.
4. Writing. The paper is well-written overall, with clear and well-designed figures and tables. The comprehensive appendix offers additional details beyond those covered in the main text. I like the way authors start their paper with an illustration on ''what makes a maze look like a maze'', which provide a easy-understanding approach for readers, especially for the concept of visual abstraction.

**Weaknesses:**

1. Size of Benchmark: one concern I have for the benchmark is relatively small size of the data. VAB only comprised 12 abstract concepts acroos 4 categoires, I know the data collection can be challenging but the limited size can decrease the significance of the benchmark evaluation. The author mentions the benchmark is divertse in terms of subject, environments and domains (Line 377). A dataset statistic over these metadata can be really helpful to support the argument.
2. Error grounding: Perception is also a challenge issue in the whole framework. It is possible that VLM have the understanding of the abstraction concept but fails to percept the image well and ground the concept within actual images (especially for images containing multiple subjects, image of cell in Figure 3).  That may also be the reasoning the schema-based framework become most helpful on couting problem. A further experiment showing VLMs can percept the image correctly is necessary.

**Questions:**

Overall the limited size of benchmark is my primary concern. I like how paper design the framework and benchmark. The benchmark provide a  good practice on handle and evaluate abstract concept, which is helpful in current literature.

Questions:
1. In llnes 399 to 400, it seems that there are multitypes of question and also each question type has different baseline accuracy (0.5 for binary in ideal). Do each categorie has a same distribution of different questions types? Otherwise the comparison over each category might be misleading.
2. Is it possible the schema generated from LLMs doesn't fit well with image? For example, the first maze image (candy maze) in Figure 5, it is unclear for human what is the layout and where is the entry point.  How does the framework handle this scenario where the generated schema bring additional concept and may cause hallucinations.



Suggestion:
1. To make reader understanding the significance of abstraction, author can enrich the introduction with some practical application of abstraction elaborate more on how human use abstraction, such abstraction enable human mind to learning from small, obtain different levels of abstraction which can later be used to trim the learning space for new concepts.

---

> ### Author Response · Authors · 2024-11-21
> **Response to Reviewer 8h5R (part 1)**
>
> We thank you for the constructive feedback!
>
> **Q. Quality of generated schemas.**
>
> A. Thank you for the suggestion! For evaluating the quality of generated schema, we added a human study using Prolific, with 20 participants. Each participant was asked 3 questions about each schema. The questions are as follow for an example concept maze:
> - Here you are given a concept, maze, and the core components underlying it: layout, walls, entry−exit. How well do these core components represent this concept on a scale of 1-7?
> - Here you are given a concept, maze, and the core components underlying it: layout, walls, entry−exit. How many components out of the ones listed accurately represent part of this concept?
> - Here you are given a concept, maze, and the core components underlying it: layout, walls, entry−exit. How many more components, if any, would you add to accurately represent this concept?
>
> We collect 3 responses for each of 12 concepts with 20 responses each, and yield a total of 720 human annotated answers.
>
> |     Survey results averaged across 20 people | How well components in the schema represent the concept (1-7) | How many components accurately represent the concept (# out of 3) | How many more components to add to accurately represent the concept (#) |
> |------------------------------------------------|-----------------------------------------------------------------|---------------------------------------------------------------------|---------------------------------------------------------------------------|
> | Schemas                                        | 5.74                                                            | 2.59                                                                | 1.59                                                                      |
>
> We see here that (a) participants rated schemas as reasonably representative, (b) 86.3% of the components were considered accurate by participants, and (c) participants suggested adding ~1.6 components on average per schema. The concept that ranks highest for the first question is the concept cell with a rating of 6.15, and lowest as the concept negotiating with 4.50. The concept solar system had the highest number of components to keep at 95%, while the concept negotiating is the lowest again at 61.7%. The concept tic-tac-toe was the most complete, with an average of 1 concept that participants might add, while the concept solar system had an average of 2.65. Though there is still room for improvement, the schemas generated by the LLM are largely accurate and aligned with human expectations. Importantly, none of the schemas were judged as fully invalid, showing the robustness of DSG’s approach. We added a detailed discussion of this human study in Appendix Section C.3.
>
> To address cases where the generated schema introduces additional components not present in an image, DSG relies on the VLM to correctly handle absence rather than hallucinate. For instance, given image 9 in the concept setting up table for two, the grounded DSG schema correctly identifies the absence of two chairs in the scene: {table: dining table, two chairs: absent, table setting: intimate dinner setup}. The VLM then incorporates this absence into its reasoning through the final prompt: “Imagine that the image represents setting up table for two, and the table is a dining table, and the two chairs are absent, and the table setting is an intimate dinner setup. What is the configuration of the seats?” Here, DSG answers correctly  with "across".
>
> To conclude, we see that the generated schemas are generally correct, and for cases where specific schema components do not exist in the given image instantiation, we rely on VLMs to detect absence in the image instead of hallucinating.

---

> > ### Author Response · Authors · 2024-11-21
> > **Response to Reviewer 8h5R (part 2)**
> >
> > **Q. Diversity of the benchmark.**
> >
> > A. Thank you for raising this point. We highlight that while our benchmark includes 12 abstract concepts, it comprises 540 image examples covering a wide range of subjects, environments, and domains. To further quantify this as per your suggestion, we used an LLM to analyze the diversity of answer choices across three dimensions: subjects, environments, and domains. The results are as follows:
> >
> > Subject types: 17 ['colors','materials', 'shapes and geometric objects', 'food items', 'fruits and vegetables',  'sweets and confections', 'furniture and household items', 'clothing and accessories', 'natural elements', 'toys and playthings', 'stationery and tools', 'building elements', 'metals', 'seasons and events', 'abstract terms', 'animal-related', 'body parts and marks']
> >
> > Environment types: 9 ['kitchen/dining', 'nature/outdoor', 'living room/bedroom', 'art/crafting room', 'playground/park', 'holiday/festive settings', 'office/study room', 'construction site', 'laboratory' ]
> >
> > Domain types: 17 ['food', 'colors', 'materials and textures', 'furniture and structures', 'nature', 'toys and games', 'educational', 'household items', 'fashion and accessories', 'seasonal and festive', 'tools and equipment', 'shapes and symbols', 'numbers and quantities', 'art and craft', 'animals and plants', 'weather and environment', 'symbols and markings']
> >
> > We see that the Visual Abstractions Benchmark covers a diverse set of images, even within the same abstract concept. We additionally highlight that our benchmark was manually curated such that each image is (a) faithful to the given concept, (b) diverse compared to other chosen images, (c) real-world, and (d) high-quality. Such a benchmark is difficult to procedurally create and filter. As a future direction, we are excited about the possibility of creating a more general benchmark, by finding a larger set of concepts (potentially querying human preferences of abstractions), scraping images from the Internet, and conducting a large-scale human study to choose and annotate suitable images. We leave this exploration for future work.
> >
> > **Q. Grounding errors.**
> >
> > A. Thank you for highlighting this point! We agree that perception is a very difficult challenge, and we note evaluating perception errors comprehensively is challenging due to the lack of ground truth annotations for all possible components in every image. We empirically show success and failure cases in Appendix Section A.6, with instances where the VLM correctly grounds concepts and where it struggles.
> >
> > **Q. Question categories.**
> >
> > A. Different types of questions exist within each concept and category. Below, we summarize the distribution of question types for each concept, where O represents open-ended questions, B represents binary questions, and C represents counting questions:
> > - tic tac toe: O, O, O
> > - maze: O, O, B
> > - treasure map: O, O, B
> > - solar system: O, B, B
> > - atom: O, O, O
> > - cell: O, O, B
> > - helping: B, B, C
> > - deceiving: O, B, C
> > - negotiating: O, B, C
> > - setting up table for two: C, C, B
> > - tidying up guest room: C, B, B
> > - putting up decorations on door: O, B, B
> >
> > We note that we do not compare performance across different categories (e.g., strategic vs science), but compare different models’ performance across categories (e.g., DSG demonstrates a strong boost in accuracy of 15% relative improvement compared to GPT-4o for strategic concepts).
> >
> > **Q. Practical application of abstraction in introduction.**
> >
> > A. Thank you for the feedback! We have edited our introduction in Section 1 accordingly.
> >
> > We thank you for all the comments, which have greatly improved our paper.

---

> > > ### Comment · Reviewer_8h5R · 2024-11-27
> > >
> > > I am pretty satisfied with the author's response to my question and will keep my rating. For the final draft, it will be better to introduce the details presented in the rebuttal to provide a concrete story.
> > >
> > > Meanwhile, as I am still concerned about the dataset, I suggest the author explore more candidates by finding ideas from dad jokes, riddles, or brainteasers. Those data contain creative thinking, analogical reasoning and abstract reasoning, which might be a good candidate for enriching the dataset.

---

> > > > ### Author Response · Authors · 2024-11-27
> > > > **Thanks to Reviewer 8h5R**
> > > >
> > > > Thank you once again for your insightful feedback, which has greatly improved the quality of our paper. We will incorporate these suggestions into our final draft.

---

### Official Review · Reviewer_XbBJ · 2024-10-28

**Soundness:** 3
**Presentation:** 3
**Contribution:** 2
**Rating:** 6
**Confidence:** 3

**Summary:**

The paper proposes the Deep Schema Grounding (DSG) framework, aiming to enhance VLM’s capability in understanding abstract visual concepts like “the maze”. The framework integrates schema representations with hierarchical grounding to guarantee the VLM’s to better capture human-aligned visual abstractions. DSG operates by extracting schemas of abstract concepts, then grounding schemas on images following a hierarchical manner, then augmenting the VQA problem with resolved schemas, resulting in improved visual reasoning. The authors also introduce the Visual Abstractions Benchmark (VAB) to systematically assess abstract concept reasoning. Experimental results indicate that DSG consistently outperforms baseline models across diverse abstract concept categories and question types.

**Strengths:**

1) The paper is well-written, it effectively conveys its proposed framwork and the motivation behinds it with clarity.
2) The experiments are well-made and the experimental analysis is comprehensive. Extensive experiments validate the effectiveness and efficiency of the DSG, which serves as a plug-and-play trick to enhance the performance of existing VLMs.
3) The pipeline that utilize schema representations with hierarchical grounding for visual concept understanding seems reasonable to the reviewer, as it emulates the human cognitive process of understanding abstract concepts.
4) The proposed method is simple yet effective, and it is easy to follow.

**Weaknesses:**

1) The gerealibility is somehow resricted. The method has been tested on only 12 types of abstract concepts and it has not been tested on the problems which invovles multiple abstract concept, which raises questions about its generalizability to a broader range of abstractions.
2) The techinical contribution of DSG is limited. The main technical contribution of this paper lies in the combination of neural-symbolic and VLM by the development of specifically designed prompts. (Prompt for LLM to generate schema, prompt for VLM to generate concept grounding). The improvements made to the prompts are relatively straightforward, and the utilization of CoT is a standard practice.
3) The risk of the hallucination of LLM/VLM. During schema construction and hierarchical prompting, the LLM may produce hallucinated errors. If errors occur in hierarchical content, individual instances may fail; however, if schema generation is flawed, it could lead to failures across an entire category of problems. The paper lacks a thorough analysis of failure cases. The authors provide some failure cases demonstration in Appendix A.3. But it seems that these results are not enough. See question 5) for further suggestion.

**Questions:**

1) Is the proposed DSG method applicable to other general visual reasoning benchmarks, such as RAVEN, Bongard, CVR, and SVRT?
2) Can DSG apply to the scenario where there are multiple abstract concepts in an image? I expect the authors to provide some insight for this, but I don't expect the authors to scale up to more experiments in the rebuttal though.
3) In table 3, the performance of schema plus grounding surpluses the performance of DSG in counting problem. Why would this happen?
4) I also have some questions regarding the length control of the schema. In the prompt provided in the appendix, it is limited to 'Keep the program simple with four or fewer components.' Why is this limitation imposed? Can the authors provide experimental results for schemas with different numbers of components? Or should the depth of the schema increase for more abstract concepts, while simpler schemas can suffice for more general concepts?
5) How robust is the method in general to failures: How often does the LLMs fail to provide a valid schema and how often does VLMs fail to provide a correct grounding? Does an invalid schema or grounding result in the failure when generate the answer?

**Details Of Ethics Concerns:**

Nothing

---

> ### Author Response · Authors · 2024-11-21
> **Response to Reviewer XbBJ (part 1)**
>
> We thank you for the constructive feedback!
>
> **Q. Experiment results for schemas with different numbers of components.**
>
> A. Thank you for bringing up this point! We agree that exploring schemas of different levels of complexity is important. Currently, our “knob” for the complexity of the generated schema is the number of components in the example prompt. Our prompt is simple, as we note potential failures of current VLMs in handling such detailed concepts, especially more spatially-focused ones (See discussion in Section 6). We hope that as LLMs become less dependent on the form of the initial prompt, they can decide the schema without being influenced by the bias in the example, and become more accurate in answering more detailed components.
>
> To explore the impact of schema complexity, we conducted an additional experiment varying the numbers of components in the example prompt: 3, 5, 7, and 9. The example prompts (all based on the same abstract concept academia expanded to different levels of detail) can be found in the added Appendix Section A.5.
>
> |                | All   | Counting  | Binary  | Open     |
> |----------------|-------|-----------|---------|----------|
> | DSG w/ 3 comp. | 0.730 | 0.689     | 0.700   | 0.771    |
> | DSG w/ 5 comp. | 0.691 | 0.622     | 0.671   | 0.733    |
> | DSG w/ 7 comp. | 0.685 | 0.667     | 0.662   | 0.713    |
> | DSG w/ 9 comp. | 0.676 | 0.700     | 0.657   | 0.683    |
>
> The results demonstrate that accuracy decreases as the number of components in the example prompt increases, likely due to current VLMs struggling with grounding more specific components. We note that all DSG variants still outperform the GPT-4o baseline. Interestingly, our findings also highlight a tendency for LLMs to generate shorter schemas, even when exposed to more detailed example prompts. Here we show the number components in the prompts and the mean number of components in the resulting schemas, averaged over all abstract concepts.
> - Prompt # 3: resulting average 3.00
> - Prompt # 5: resulting average 4.66
> - Prompt # 7: resulting average 5.75
> - Prompt # 9: resulting average 7.00
>
> Notably, although the number of components in the example prompt does affect the average number of components in resulting schemas, this effect is less prominent and the resulting average plateaus. This phenomenon shows that schemas may potentially converge to shorter lengths.
> We agree that indeed abstract concepts should be deeper and have more components, and vice versa. We note though current VLM limitations constrain schema complexity, as these capabilities improve, DSG’s performance is expected to scale, enabling us to adopt more detailed schemas without sacrificing effectiveness. Thanks to your feedback, we have included a more in-depth discussion of this tradeoff and added the ablation study results to Appendix Section A.5.

---

> > ### Author Response · Authors · 2024-11-21
> > **Response to Reviewer XbBJ (part 2)**
> >
> > **Q. Evaluation of generated schemas and grounding results.**
> >
> > A. Thank you for the suggestion! For evaluating the validity of generated schema, we added a human study using Prolific, with 20 participants. Each participant was asked 3 questions about each schema. The questions are as follow for an example concept maze:
> > - Here you are given a concept, maze, and the core components underlying it: layout, walls, entry−exit. How well do these core components represent this concept on a scale of 1-7?
> > - Here you are given a concept, maze, and the core components underlying it: layout, walls, entry−exit. How many components out of the ones listed accurately represent part of this concept?
> > - Here you are given a concept, maze, and the core components underlying it: layout, walls, entry−exit. How many more components, if any, would you add to accurately represent this concept?
> >
> > We collect 3 responses for each of 12 concepts with 20 responses each, and yield a total of 720 human annotated answers.
> >
> > |     Survey results averaged across 20 people | How well components in the schema represent the concept (1-7) | How many components accurately represent the concept (# out of 3) | How many more components to add to accurately represent the concept (#) |
> > |------------------------------------------------|-----------------------------------------------------------------|---------------------------------------------------------------------|---------------------------------------------------------------------------|
> > | Schemas                                        | 5.74                                                            | 2.59                                                                | 1.59                                                                      |
> >
> > We see here that (a) participants rated schemas as reasonably representative, (b) 86.3% of the components were considered accurate by participants, and (c) participants suggested adding ~1.6 components on average per schema. The concept that ranks highest for the first question is the concept cell with a rating of 6.15, and lowest as the concept negotiating with 4.50. The concept solar system had the highest number of components to keep at 95%, while the concept negotiating is the lowest again at 61.7%. The concept tic-tac-toe was the most complete, with an average of 1 concept that participants might add, while the concept solar system had an average of 2.65. Though there is still room for improvement, the schemas generated by the LLM are largely accurate and aligned with human expectations. Importantly, none of the schemas were judged as fully invalid, showing the robustness of DSG’s approach. We added a detailed discussion of this human study in Appendix Section C.3.
> >
> > We also agree that grounding is a very important and difficult challenge, and we note that it’s challenging to evaluate as there is no ground truth annotation for all possible components in every image. We show failure cases in Appendix Section A.6. Empirically, errors in the grounding stage may lead to errors when generating the final answer, but VLMs have the ability to self-correct for most cases, especially cases where the downstream query does not directly ask about an incorrect schema component. We believe DSG’s performance will scale naturally as VLMs improve in performance.

---

> > > ### Author Response · Authors · 2024-11-21
> > > **Response to Reviewer XbBJ (part 3)**
> > >
> > > **Q. Contribution of DSG.**
> > >
> > > A. We note that DSG’s contribution lies in its simplicity and generality—it can be applied to any combination of LLM and VLM, making it broadly applicable. Moreover, we note that the key idea behind DSG precisely lies in schema decomposition (e.g., in the form of extracting components: layout, walls, and entry and exit) to explicitly guide the VLM to interpret the scene in a structured way. We note that the specific method of processing these questions—whether in a single CoT query [1] or across multiple queries—is not fundamentally important. While in theory, these schema reasoning steps could be merged into a single CoT-style prompt, our experiments show that multi-round execution with explicit grounding steps significantly improves performance in practice.
> > >
> > > For example, in Table 3 of the main text, the first row of the ablation shows results when the VLM is given the hierarchical schema for reasoning in a single query, without using the explicit grounding steps. In addition, we added an experiment listing out each component and asking the model to think step-by-step: “Imagine that the image represents a maze. Think step-by-step about recognizing the layout, walls, entry exit one-by-one. What is the player in this maze?”. We report results below, and compare (1) GPT-4o, (2) the single-round CoT baseline with schema components, (3) the single-round CoT baseline with the full hierarchical schema, and (4) our full multi-round DSG framework. All results are averaged over 5 runs.
> > >
> > > |                                            | All   | Counting  | Binary  | Open     |
> > > |--------------------------------------------|-------|-----------|---------|----------|
> > > | GPT-4o                                     | 0.664 | 0.604     | 0.655   | 0.693    |
> > > | Single-round with schema components        | 0.667 | 0.607     | 0.661   | 0.696    |
> > > | Single-round with hierarchical    schema   | 0.681 | 0.629     | 0.672   | 0.709    |
> > > | Multi-round with hierarchical schema (DSG) | 0.730  | 0.704     | 0.690    | 0.776    |
> > >
> > > We see that DSG’s multi-round execution outperforms single-round, CoT-style baselines, likely due to its explicit guidance and grounding. Thanks to your feedback, we’ve added these experiment results in Appendix Section A.3.
> > >
> > > [1] Wei, Jason, et al. "Chain-of-thought Prompting Elicits Reasoning in Large Language Models." Advances in Neural Information Processing Systems 35 (2022): 24824-24837.
> > >
> > > **Q. Multiple abstract concepts in image.**
> > >
> > > A. In theory, DSG can indeed handle scenarios with multiple abstract concepts in an image.
> > > Concretely, we can use schemas from multiple abstract concepts as context to the VLM, by simply grounding schema components from each concept, then chaining together the grounded schemas. However, it’s difficult to curate a dataset, or find many real-world examples where multiple abstractions coexist meaningfully within a single image. Such a dataset would require careful design to ensure the concepts interact in non-trivial ways, rather than existing as isolated components. While this is an exciting direction, it is beyond the scope of our current work.
> > >
> > > **Q. Applicability to other benchmarks.**
> > >
> > > A. DSG is currently designed for understanding abstract visual concepts, and not for benchmarks in which there are no underlying abstract concepts, or for scenarios in which the concepts need to be discovered. Therefore, it is not directly applicable to benchmarks like RAVEN, Bongard, CVR, and SVRT, where the abstraction often needs to be found as part of the reasoning process. Extending DSG to include a discovery stage for underlying abstractions is an exciting direction for future work. This could involve incorporating methods to identify abstract patterns dynamically, which could make DSG applicable to a broader range of visual reasoning benchmarks.
> > >
> > > **Q. Performance of schema with grounding compared to DSG in counting problems.**
> > >
> > > A. Thank you for pointing this out! We hypothesize that for counting problems, the key challenge is in correctly identifying the individual components being counted. In such cases, a model with sequential and explicit grounding directly focuses on these components, which may be sufficient for accurate reasoning. The hierarchical structure and context history provided by DSG, while beneficial for more complex tasks, may not be helpful for simpler counting tasks.
> > >
> > > We thank you for all the comments, which have greatly improved our paper.

---

> > > > ### Comment · Reviewer_XbBJ · 2024-11-27
> > > >
> > > > I have read the rebuttal carefully and most of my concerns are addressed. Hence, I decide to keep my original score and am leaning to accept this paper.

---

> > > > > ### Author Response · Authors · 2024-11-27
> > > > > **Thanks to Reviewer XbBJ**
> > > > >
> > > > > Thank you once again for your insightful feedback, which has greatly improved the quality of our paper.

---

### Official Review · Reviewer_nf1E · 2024-11-04

**Soundness:** 4
**Presentation:** 4
**Contribution:** 3
**Rating:** 8
**Confidence:** 4

**Summary:**

This paper introduces a VQA benchmark and method for visual abstract reasoning.

The "Visual Abstractions Benchmark" consists of 180 images, annotated with 3 questions each (and each question has 5 answer annotations from Prolific workers). These questions ask about abstract concepts in the image: for example, "Q: Imagine that the image represents a maze. Are there clear entrances or exits in the maze? [Yes or No]" (Figure 1; an image of an ice skating rink partitioned like a maze).

The method, "Deep Schema Grounding (DSG)", is a prompting framework that (1) uses an LLM to decompose an abstract concept into a schema, (2) uses a VLM to ground the "primitive" concepts from the schema to concrete concepts in the image, and (3) uses a VLM to answer the question given the schema and grounding.
1. The schema is a DAG: a "maze" would decompose into "layout" and "walls", and "layout" further decomposes into "entry-exit".
2. A VQA question is used for grounding: e.g. given the image, "..., what is the layout?". This graph is hierarchically grounded: e.g. "entry-exit" is grounded using the question "..., the layout is [], what is the entry and exit?". The ice rink example above entails the following grounding: "{layout: grid-like, walls: ice, entry-exit: gates}".
3. The image and grounded schema are joined as a VQA question for the VLM to produce a final answer.

Applying this framework improves performance of GPT-4o by 10% (relative) on the benchmark.

**Strengths:**

This paper is extremely clear and straightforward.

In general, I believe the comparisons are fair and sufficient ablations are provided, so it is methodologically sound. For example, they show a 10% relative improvement when applying their method with GPT-4o and 7% with (the open-source and weaker) LLaVA model. They also compare against other methods (ViperGPT and VisProg) that also use GPT models (GPT-3) for visual question decomposition, but don't perform nearly as well on this benchmark. Finally, they provide sufficient ablations of their own method (separating the schema/grounding/hierarchy/context components and showing incremental gains of each).

I believe this is a creative benchmark and that existing vision–language benchmarks under-explore abstract visual concepts. The method is task-specific, but I think that is fine, because it is prompting-based. I don't think the approach is entirely original but I think it sufficiently rounds out the paper: they show how explicit structure improves performance of existing models on their benchmark.

Altogether: this paper presents a benchmark, shows that existing vision–language models underperform, and introduces an explicit prompting method that can be applied with any LLM/VLM to improve performance. The best performance with this method is still just 73% (with the strongest, closed models) or 44% (with open-source models), leaving sufficient challenge for future work.

**Weaknesses:**

I believe the construction of the dataset is under-specified (and I also checked the appendix). I understand that answer annotations were provided by crowd workers. However, how were the questions written? How were the images selected (and where do they come from: L376 just says "the Internet")? Likewise for the 12 abstract concepts and their 4 categories. Right now, I am imagining that the authors curated these themselves. That could be fine, but I would like to know more details, so that we can determine whether there may be any biases: e.g. does the introduced method perform better on this particular selection of abstract concepts?

I notice that answers in the open-ended setting are evaluated using BERTScore. However, (Kamalloo 2023) says

> unsupervised semantic similarity-based evaluation metrics such as BERTScore (Zhang et al., 2020) that rely on token-level matching of contextualized representations exhibit poor correlation with human judgment in QA evaluation (Chen et al., 2019) and lack the ability to capture attributability (Maynez et al., 2020)

I also find this to be true in general. Could you show examples, so we can see if the metric is indeed suitable for this distribution of data? Otherwise, could you consider using a method that is better aligned with human judgments for this evaluation: possibly (Kamalloo 2023)?

Reference: [Evaluating Open-Domain Question Answering in the Era of Large Language Models](https://aclanthology.org/2023.acl-long.307) (Kamalloo 2023)

**Questions:**

The word "lifted" is used in several places (e.g. "lifted symbols" or "lifted rules"). I think this terminology isn't entirely obvious. If this has a specific meaning, could the authors please elaborate in text (or just use a different word if not)?

It could be informative to clarify the distribution of schemas based on their graph depth in the text. E.g. I see that "cell" has depth 2 and "atom" has depth 3 (L875-881). The single in-context prompt for schema generation is "academia" (depth=3; Sec. C.1). Could abstract concepts exist with depths larger than 3? Would the LLM limit their decompositions to depth=3 (at most) because that is the only prompt? I think the authors could elaborate further on their prompt engineering and exploration, especially since their method is a prompting framework.

---

> ### Author Response · Authors · 2024-11-21
> **Response to Reviewer nf1E (part 1)**
>
> We thank you for the constructive feedback!
>
> **Q. Complexity of schemas.**
>
> A. Thank you for bringing up this point! We agree that exploring schemas of different levels of complexity is important. Currently, our “knob” for the complexity of the generated schema is the number of components in the example prompt. Our prompt is simple, as we note potential failures of current VLMs in handling such detailed concepts, especially more spatially-focused ones (See discussion in Section 6). We hope that as LLMs become less dependent on the form of the initial prompt, they can decide the schema without being influenced by the bias in the example.
>
> To explore the impact of schema complexity, we conducted an additional experiment varying the numbers of components in the example prompt: 3, 5, 7, and 9. The example prompts (all based on the same abstract concept academia expanded to different levels of detail) can be found in the added Appendix Section A.5.
>
> |                | All   | Counting  | Binary  | Open     |
> |----------------|-------|-----------|---------|----------|
> | DSG w/ 3 comp. | 0.730 | 0.689     | 0.700   | 0.771    |
> | DSG w/ 5 comp. | 0.691 | 0.622     | 0.671   | 0.733    |
> | DSG w/ 7 comp. | 0.685 | 0.667     | 0.662   | 0.713    |
> | DSG w/ 9 comp. | 0.676 | 0.700     | 0.657   | 0.683    |
>
> The results demonstrate that accuracy decreases as the number of components in the example prompt increases, likely due to current VLMs struggling with grounding more specific components. We note that all DSG variants still outperform the GPT-4o baseline. Interestingly, our findings also highlight a tendency for LLMs to generate shorter schemas, even when exposed to more detailed example prompts. Here we show the number components in the prompts and the mean number of components in the resulting schemas, averaged over all abstract concepts.
> - Prompt # 3: resulting average 3.00
> - Prompt # 5: resulting average 4.66
> - Prompt # 7: resulting average 5.75
> - Prompt # 9: resulting average 7.00
>
> Notably, although the number of components in the example prompt does affect the average number of components in resulting schemas, this effect is less prominent and the resulting average plateaus. This phenomenon shows that schemas may potentially converge to shorter lengths.
>
> We note that current VLM limitations constrain schema complexity. However, as these capabilities improve, DSG’s performance is expected to scale, enabling us to adopt more detailed schemas without sacrificing effectiveness. Thanks to your feedback, we have included a more in-depth discussion of this tradeoff and added the ablation study results to Appendix Section A.5.
>
> **Q. Free-response evaluation with Kamalloo’s metric.**
>
> A. Thanks for pointing us to this reference! Below, we report free-response results by  incorporating one of Kamalloo’s proposed strategies for evaluating free-response answers. In particular, we use zero-shot evaluation via LLM prompting, which includes the ground truth and predicted answer in the prompt, similar to the semantic similarity mechanism, to verify the correctness of the prediction. We use GPT-4 as the base LLM, and compare GPT-4o with DSG.
>
> |        | All   | Counting | Binary | Open     |
> |--------|-------|----------|--------|----------|
> | GPT-4o | 36.67 | 45.56    | 55.24  | 14.58    |
> | DSG    | 37.78 | 53.33    | 49.52  | 21.25    |
>
> We see that DSG outperforms GPT-4o overall, with particularly strong results on counting and open-ended questions, and shows stronger signal than with BERTScores. However, performance on binary questions is lower, which we hypothesize is due to some of the components being potentially misleading or ambiguous in binary contexts. We added these evaluation results in Appendix Section A.2.

---

> > ### Author Response · Authors · 2024-11-21
> > **Response to Reviewer nf1E (part 2)**
> >
> > **Q. Dataset construction.**
> >
> > A. Thank you for this feedback! We note that the first concepts we explored were strategic games, including mazes and tic-tac-toe. We then expanded the categories to create a more diverse benchmark, inspired by areas where abstract concepts would occur in the real-world, including scientific concepts where many classroom science projects exist, social concepts with theory-of-mind examples, and domestic concepts with tasks that do not have predefined and exact configurations of end states. Images were manually curated from the Internet, primarily using Google Image Search. We searched for diverse real-world representations of each concept (e.g., tic-tac-toe on the beach). Images were chosen such that each image is (a) faithful to the given concept, (b) diverse compared to other chosen images, (c) real-world, and (d) high-quality; this process is difficult to procedurally replicate. We wrote general questions for each concept, aiming for broad applicability and alignment with the images. These questions focus on reasoning grounded in the abstract concept rather than specific visual details of individual images. Once images and questions were fixed, we sent them to human annotators via Prolific, before testing our method on the finalized dataset. The concepts in the benchmark were not changed. We have added these benchmark construction details to Appendix Section B.2.
> >
> > **Q. Clarification on the word “lifted”.**
> >
> > A. Thanks for pointing this out! We clarify that we use “lifted” in direct contrast to “grounded” concepts, and we use it to refer to symbols or rules that are not tied to any specific downstream visual input or instance (e.g., any tic-tac-toe game has a 3x3 grid vs. tic-tac-toe in image29.jpg has a 3x3 grid). For example, a “lifted symbol” represents a general concept that can be applied across various scenarios, whereas a “grounded symbol” is instantiated with specific visual details. We updated our introduction in Section 1 to clarify.
> >
> > We thank you for all the comments, which have greatly improved our paper.

---

> > > ### Comment · Reviewer_nf1E · 2024-11-23
> > >
> > > Thanks so much to the authors for their response and clarifications. I appreciate the extra effort in showing results on a different metric. I have no remaining concerns. I will keep my score of 8, which I feel is/was appropriately strong :)

---

> > > > ### Author Response · Authors · 2024-11-26
> > > > **Thanks to Reviewer nf1E**
> > > >
> > > > Thank you once again for your insightful feedback, which has greatly improved the quality of our paper.

---

### Official Review · Reviewer_P1iE · 2024-11-08

**Soundness:** 3
**Presentation:** 3
**Contribution:** 2
**Rating:** 6
**Confidence:** 3

**Summary:**

The paper proposes Deep Schema Grounding (DSG), a method to break down a visual concept into smaller concepts with dependencies on other concepts. The authors show the capability of DSG in solving complex visual question answering task. A visual abstraction benchmark is proposed with 12 abstract concepts and 180 images.

**Strengths:**

The paper provides a good solution for VLMs to better understand abstract concepts in an image. The presentation of the paper is clear with many figures for demonstration. The experiments are comprehensive, making the main message convincing. The benchmark created is novel and interesting.

**Weaknesses:**

In terms of the idea behind DSG, it seems to be close to chain of thought with specific instructions. For example, the maze example in the paper can be integrated with just one prompt: "Imagine that the image represents a maze. <the question> Think step by step by recognizing the layout of the maze, the walls of the maze, then the entry and exit of the maze one by one." Maybe gpt-4o can automatically do this even without the instructions. This is probably why in Table 7 if the generation is free form, DSG does not outperform gpt-4o much despite using more API calls.

Therefore, I am a bit concerned whether these types of schemas are necessary. After all, the dependency graph is generated by gpt-4, so at least gpt-4o should know how to do it if prompted well. But I agree that for smaller models, DSG can be very useful.

Another concern I have is about the diversity of the benchmark because the number of categories is quite limited.

**Questions:**

Could you try something like chain of thought prompting as I mentioned above?

Is there any way to generalize the idea to create a more general benchmark?

---

> ### Author Response · Authors · 2024-11-21
> **Response to Reviewer Reviewer P1iE (part 1)**
>
> We thank you for the constructive feedback!
>
> **Q. Difference between DSG and chain-of-thought.**
>
> A. The key idea behind DSG precisely lies in schema decomposition (e.g., in the form of extracting components: layout, walls, and entry and exit) to guide the VLM to interpret the scene. DSG introduces an explicit guidance to the VLM system, by first generating a decomposition of the target concept, parsing the image, then answering the question. We note that the specific method of processing these questions—whether in a single query or across multiple queries—is not fundamentally important. While in theory, these steps could be merged into a single CoT-style prompt, our experiments show that multi-round execution with explicit grounding steps significantly improves performance in practice.
>
> For example, in Table 3 of the main text, the first row of the ablation shows results when the VLM is given the hierarchical schema for reasoning in a single query, without using the explicit grounding steps. In addition, we ran an experiment based on your suggested prompt, listing out each component and asking the model to think step-by-step: “Imagine that the image represents a maze. Think step-by-step about recognizing the layout, walls, entry exit one-by-one. What is the player in this maze?”. We report results below, and compare (1) GPT-4o, (2) the suggested single-round CoT baseline with schema components, (3) the single-round CoT baseline with the full hierarchical schema, and (4) our full multi-round DSG framework. All results are averaged over 5 runs.
>
> |                                            | All   | Counting  | Binary  | Open     |
> |--------------------------------------------|-------|-----------|---------|----------|
> | GPT-4o                                     | 0.664 | 0.604     | 0.655   | 0.693    |
> | Single-round with schema components        | 0.667 | 0.607     | 0.661   | 0.696    |
> | Single-round with hierarchical    schema   | 0.681 | 0.629     | 0.672   | 0.709    |
> | Multi-round with hierarchical schema (DSG) | 0.730  | 0.704     | 0.690    | 0.776    |
>
> We see that DSG’s multi-round execution outperforms single-round, CoT-style baselines, likely due to its explicit guidance and grounding.
>
> We agree that DSG’s schema decomposition is especially helpful for smaller models, which may lack the reasoning capabilities of larger models like GPT-4o. Thanks to your feedback, we’ve clarified our ablation setup in the updated Section 5 of the main text, and added these experiment results in Appendix Section A.3.
>
> [1] Wei, Jason, et al. "Chain-of-thought Prompting Elicits Reasoning in Large Language Models." Advances in Neural Information Processing Systems 35 (2022): 24824-24837.

---

> > ### Author Response · Authors · 2024-11-21
> > **Response to Reviewer Reviewer P1iE (part 2)**
> >
> > **Q. Diversity of the benchmark and ways to create a more general benchmark.**
> >
> > A. Thank you for raising this point. We highlight that while our benchmark includes 12 abstract concepts, it comprises 540 image examples covering a wide range of subjects, environments, and domains. To further quantify this, we used an LLM to analyze the diversity of answer choices across three dimensions: subjects, environments, and domains. The results are as follows:
> >
> > Subject types: 17 ['colors','materials', 'shapes and geometric objects', 'food items', 'fruits and vegetables',  'sweets and confections', 'furniture and household items', 'clothing and accessories', 'natural elements', 'toys and playthings', 'stationery and tools', 'building elements', 'metals', 'seasons and events', 'abstract terms', 'animal-related', 'body parts and marks']
> >
> > Environment types: 9 ['kitchen/dining', 'nature/outdoor', 'living room/bedroom', 'art/crafting room', 'playground/park', 'holiday/festive settings', 'office/study room', 'construction site', 'laboratory' ]
> >
> > Domain types: 17 ['food', 'colors', 'materials and textures', 'furniture and structures', 'nature', 'toys and games', 'educational', 'household items', 'fashion and accessories', 'seasonal and festive', 'tools and equipment', 'shapes and symbols', 'numbers and quantities', 'art and craft', 'animals and plants', 'weather and environment', 'symbols and markings']
> >
> > We see that the Visual Abstractions Benchmark covers a diverse set of images, even within the same abstract concept. We additionally highlight that our benchmark was manually curated such that each image is (a) faithful to the given concept, (b) diverse compared to other chosen images, (c) real-world, and (d) high-quality. Such a benchmark is difficult to procedurally create and filter. As a future direction, we are excited about the possibility of creating a more general benchmark, by finding a larger set of concepts (potentially querying human preferences of abstractions), scraping images from the Internet, and conducting a large-scale human study to choose and annotate suitable images. We leave this exploration for future work.
> >
> > We thank you for all the comments, which have greatly improved our paper.

---

> ### Comment · Reviewer_P1iE · 2024-11-26
>
> Thanks for the authors' response, I have increased my score accordingly.

---

> > ### Author Response · Authors · 2024-11-26
> > **Thanks to Reviewer P1iE**
> >
> > Thank you once again for your insightful feedback, which has greatly improved the quality of our paper.

---

### Official Review · Reviewer_yKhN · 2024-11-10

**Soundness:** 3
**Presentation:** 3
**Contribution:** 3
**Rating:** 6
**Confidence:** 4

**Summary:**

The paper introduces a framework called Deep Schema Grounding (DSG) and a VQA benchmark for evaluating abstract visual reasoning of VLMs. For a given image and a question about the concept in the image, DSG first generates a schema of the concept using LLM. Then it grounds the components of the schema onto images using VLMs. Once the schema is grounded, the grounding information is given as context in text from to the VML to answer the question about the image. They show that giving grounded schema context to the VLMs improves their ability to correctly answer questions about abstract concepts in the image.

**Strengths:**

- Innovative use of pre-trained LLMs and VLMs to give additional context (in terms of grounded schemas) for VQA. In principle, this additional information should help answer pre-trained VLMs to better answer the questions related to the abstract concepts in the image. (However I think that the generated schemas shown in the appendix of the paper are not detailed enough to achieve this - please check weakness section)

- Hierarchical Grounding: I like how they have used a hierarchical method of grounding components of the schemas. Grounding independent concepts first and then the dependent concepts of the schema should be the correct way for grounding which is used in the paper.

**Weaknesses:**

- Schemas are not detailed enough to give information about the concept. For example, tic-tac-toe schema include {board, symbols, strategy}. Although a tic-tac-toe game has  {board, symbols, strategy} it is not complete. Many board games have {board, symbols, strategy}. This schema does not tell that the board is a 3x3 grid. Similarly, for “negotiating” the schema is {participants, setting, object}. This schema could be of many different settings. (p.s. I do understand that the capabilities of current VLMs are not enough to handle detailed concepts)

- It is shown that giving grounded schema as additional context to VLM to answer VQA question is improving the performance. But it is not shown or discussed why?/how? is the additional context improving the performance. For instance, in the example given in figure 1 of the paper where the question asks “What is the player in the maze?”, how is giving information about the layout, walls and entry-exit helping a VLM answer question about the ‘player’.

**Questions:**

- Many recent works (https://arxiv.org/pdf/2305.10355, https://arxiv.org/pdf/2401.06209, others) have shown that VLMs  are ‘blind’ and don’t actually look at the visual information in images. VLMs mostly rely on language and questions in VQA benchmarks can be answered by having a good language prior. I was wondering how much of the questions in the proposed benchmark can be correctly answered by “language” only? It would be great to show a “language-only” baseline for the proposed VQA benchmark. Especially after adding grounded schema as context- it can show the impact of providing grounded schema too.

- A qualitative analysis of how/why is the grounded schema is improving VQA capabilities of the VLM would be great. Especially for the cases where the question does not ask anything about the concepts of the schema (like the example of “What is the player in the maze?” mentioned in the weakness section)

- Are the generated schemas verified by some expert? How are we sure that the schemas provided by the LLMs are correct for the provided concepts?

**Details Of Ethics Concerns:**

The supplementary material includes a link to the dataset. The link is: https://downloads.cs.stanford.edu/viscam/VisualAbstractionsDataset/VAD.zip

The link is not fully anonymized and may violate double-blind policy.

I don't know if this violates the double-blind policy and therefore I'm flagging it for concern.

---

> ### Author Response · Authors · 2024-11-21
> **Response to Reviewer yKhN (part 1)**
>
> We thank you for the constructive feedback!
>
> **Q. Schema detailedness and complexity.**
>
> A. Thank you for bringing up this point! We agree that there’s a tradeoff between encouraging more detailed components in a schema and avoiding both (a) components that are less broadly applicable across all image instantiations, and (b) as mentioned by the reviewer, potential failures of current VLMs in handling such detailed concepts, especially more spatially-focused ones. We note that as capabilities of VLMs increase, DSG’s performance will scale accordingly, and we can increase the detail of the schema without making this tradeoff.
>
> In addition, currently our “knob” for the complexity of the generated schema is the number of components in the example prompt. As LLMs become less dependent on the form of the initial prompt, they can decide the schema without being influenced by the bias in the example.
>
> To explore the impact of schema complexity and better understand the capabilities of current VLMs, we conducted an additional experiment varying the numbers of components in the example prompt: 3, 5, 7, and 9. The example prompts (all based on the same abstract concept academia expanded to different levels of detail) can be found in the added Appendix Section A.5.
> |                | All   | Counting  | Binary  | Open     |
> |----------------|-------|-----------|---------|----------|
> | DSG w/ 3 comp. | 0.730 | 0.689     | 0.700   | 0.771    |
> | DSG w/ 5 comp. | 0.691 | 0.622     | 0.671   | 0.733    |
> | DSG w/ 7 comp. | 0.685 | 0.667     | 0.662   | 0.713    |
> | DSG w/ 9 comp. | 0.676 | 0.700     | 0.657   | 0.683    |
>
> The results demonstrate that accuracy decreases as the number of components in the example prompt increases, likely due to current VLMs struggling with grounding more specific components. We note that all DSG variants still outperform the GPT-4o baseline. Interestingly, our findings also highlight a tendency for LLMs to generate shorter schemas, even when exposed to more detailed example prompts. Here we show the number components in the prompts and the mean number of components in the resulting schemas, averaged over all abstract concepts.
> - Prompt # 3: resulting average 3.00
> - Prompt # 5: resulting average 4.66
> - Prompt # 7: resulting average 5.75
> - Prompt # 9: resulting average 7.00
>
> Notably, although the number of components in the example prompt does affect the average number of components in resulting schemas, this effect is less prominent and the resulting average plateaus. This phenomenon shows that schemas may potentially converge to shorter lengths.
>
> We note that current VLM limitations constrain schema complexity. However, as these capabilities improve, DSG’s performance is expected to scale, enabling us to adopt more detailed schemas without sacrificing effectiveness. Thanks to your feedback, we have included a more in-depth discussion of this tradeoff and added the ablation study results to Appendix Section A.5.

---

> ### Author Response · Authors · 2024-11-21
> **Response to Reviewer yKhN (part 2)**
>
> **Q. Language-only baseline and impact of providing grounded schemas.**
>
> A. Thank you for the suggestion! We added two additional language baselines based on your suggestion. The first is a language-only baseline, where GPT-4o only has access to the question itself without input image or schema. The second is a language-with-schema baseline, where GPT-4o has access to the question and the DSG grounded schema (the schema is extracted from the image and provides context), but the final query itself does not have access to the image. We report results averaged over 5 runs below.
>
> |                               | All   | Counting  | Binary  | Open     |
> |-------------------------------|-------|-----------|---------|----------|
> | Language-only                 | 0.397 | 0.300       | 0.510    | 0.335    |
> | Language with grounded schema | 0.522 | 0.440      | 0.530    | 0.546    |
>
> Notably, the language-only baseline performs worse than all open-source and closed-source VLMs, showing that the visual input is essential for achieving higher accuracy on the benchmark. We note that the language-only baseline outperforms integrated LLMs with API baselines, as these methods tend to fail in execution.
>
> With DSG’s grounded schema, the language model performs significantly better, highlighting the importance of the holistic context provided by the grounded schema. We see that in open-ended questions, the grounded schema significantly improves performance of the language-only baseline. We hypothesize that this is because the schema helps rule out implausible answer choices by leveraging the grounded components. We add these experiment  results and discussion in Appendix Section A.4.
>
> **Q. Verification of generated schemas.**
>
> A. Thank you for raising this point! To evaluate the correctness and quality of the generated schemas, we added a human study using Prolific, with 20 participants. Each participant was asked 3 questions about each schema. The questions are as follow for an example concept maze:
> - Here you are given a concept, maze, and the core components underlying it: layout, walls, entry−exit. How well do these core components represent this concept on a scale of 1-7?
> - Here you are given a concept, maze, and the core components underlying it: layout, walls, entry−exit. How many components out of the ones listed accurately represent part of this concept?
> - Here you are given a concept, maze, and the core components underlying it: layout, walls, entry−exit. How many more components, if any, would you add to accurately represent this concept?
>
> We collect 3 responses for each of 12 concepts with 20 responses each, and yield a total of 720 human annotated answers.
>
> |     Survey results averaged across 20 people | How well components in the schema represent the concept (1-7) | How many components accurately represent the concept (# out of 3) | How many more components to add to accurately represent the concept (#) |
> |------------------------------------------------|-----------------------------------------------------------------|---------------------------------------------------------------------|---------------------------------------------------------------------------|
> | Schemas                                        | 5.74                                                            | 2.59                                                                | 1.59                                                                      |
>
> We see here that (a) participants rated schemas as reasonably representative, (b) 86.3% of the components were considered accurate by participants, and (c) participants suggested adding ~1.6 components on average per schema. The concept that ranks highest for the first question is the concept cell with a rating of 6.15, and lowest as the concept negotiating with 4.50. The concept solar system had the highest number of components to keep at 95%, while the concept negotiating is the lowest again at 61.7%. The concept tic-tac-toe was the most complete, with an average of 1 concept that participants might add, while the concept solar system had an average of 2.65. Though there is still room for improvement, the schemas generated by the LLM are largely accurate and aligned with human expectations. Importantly, none of the schemas were judged as fully invalid, showing the robustness of DSG’s approach. We added a detailed discussion of this human study in Appendix Section C.3.

---

> ### Author Response · Authors · 2024-11-21
> **Response to Reviewer yKhN (part 3)**
>
> **Q. Why grounded schemas are helpful.**
>
> A. We hypothesize that the grounded schemas provide structured, holistic context to the VLM (by way of assigning roles to entities in the image), enabling the model to organize information more effectively and improve its performance. This grounded context helps downstream reasoning, even when the query does not explicitly involve a schema component. Importantly, we note the generality of DSG, which does not rely on knowing the query a priori.
>
> Though difficult to analyze the inner workings of the VLM, to address your example, we conducted an analysis of the "What is the player in the maze?" question. Here, the grounded schema explicitly includes components "layout," "walls," and "entry-exit." When we asked the VLM (GPT-4o) to explain its reasoning while using the grounded schema, it provided the following:
>
> "Reasoning:
> The layout provides structure, which helps identify where pathways and spaces are within the maze. The walls made of branches define boundaries and paths, making it possible to understand where movement might occur within the maze. The entry and exit points indicate potential traversal through the maze. Since no identifiable player (grass, moss, stick, or anything distinct from the environment) is visible within the maze structure itself, the answer is: none."
>
> This reasoning supports our hypothesis: the grounded schema assigns semantic roles to image components, which helps the VLM disambiguate entities in the scene and improve downstream reasoning. In this case, the schema enables the VLM to interpret the absence of a distinct "player" by contextualizing the maze structure. We add this discussion in Appendix Section A.8.
>
> **Q. Dataset link.**
>
> A. Thanks for your careful feedback regarding the dataset download URL in the supplementary material! To address the concern and avoid any confusion, we have updated the link. We would also like to clarify the following points:
> - The URL is embedded for direct downloading, and readers are not expected to see the URL itself. The URL does not appear anywhere in the text of the main paper or the supplementary material.
> - The URL does not contain any personal identifiers or link to specific individuals, nor does it reveal any names.
> - Similar to how researchers might include images of a university in their work, the inclusion of a university name in a dataset URL does not necessarily identify any author or institution. Thanks!
>
> We thank you for all the comments, which have greatly improved our paper.

---

> ### Comment · Reviewer_yKhN · 2024-11-26
>
> I thank authors for clarifying my queries. I am improving my rating.

---

> > ### Author Response · Authors · 2024-11-26
> > **Thanks to Reviewer yKhN**
> >
> > Thank you once again for your insightful feedback, which has greatly improved the quality of our paper.

---

### Meta-Review · Area_Chair_1tK9 · 2024-12-20

**Metareview:**

This paper introduces Deep Schema Grounding (DSG), which enhances VLMs in abstract visual reasoning by using an LLM to generate schemas that decompose abstract concepts into smaller components, which are then grounded onto images for answering related questions. Experimental results show that DSG improves GPT-4o’s performance by 10% on the newly constructed Visual Abstractions Benchmark (VAB), outperforming other VLMs and demonstrating the contribution of each DSG component through ablation studies.

There are five reviewers for this paper. The reviewers highlight the key strengths of the paper, including its simplicity and effectiveness, along with its contributions to both the benchmark and methodology, supported by well-structured experiments and comprehensive ablation studies.

Meanwhile, the main concerns from the reviewers are 1) the dataset's limited size, along with its lack of diversity in categories and ability to handle multiple abstract concepts in a single image, may not offer enough breadth to effectively evaluate generalizability. 2) the DSG approach seems similar to chain-of-thought prompting with specific instructions. Some reviewers (Reviewer P1iE) question whether schemas are necessary, suggesting that GPT-4 could handle the task with a well-designed prompt. 3) Some schemas, such as those for tic-tac-toe and negotiating, are too generic and fail to capture key details, limiting the information provided to the VLMs. 4) There is a lack of explanation as to why or how the additional context provided by the grounded schema improves performance in VQA. 5) The DSG method has not been tested on other benchmarks, and its robustness to errors or failures in schema generation or concept grounding is unclear.

During the author-reviewer discussion phase, the authors' clarifications and additional experimental results effectively addressed the reviewers' concerns. All reviewers rated the paper positively and recommended acceptance. ACs support the reviewers’ recommendation. Authors should address the key points raised in the reviews, particularly the requests for additional analyses and experiments, when preparing the final version.

**Additional Comments On Reviewer Discussion:**

During the author-reviewer discussion phase, the authors' clarifications and additional experimental results effectively addressed the reviewers' concerns. All reviewers rated the paper positively and recommended acceptance. ACs support the reviewers’ recommendation. Authors should address the key points raised in the reviews, particularly the requests for additional analyses and experiments, when preparing the final version.

---

### Decision · Program_Chairs · 2025-01-22

Accept (Poster)